



# A new method for diagnosing effective radiative forcing from aerosol-cloud interactions in climate models

Brandon M. Duran[1], Casey J. Wall[2,3], Nicholas J. Lutsko[1], Takuro Michibata[4], Po-Lun Ma[6], Yi Qin[6], Margaret L. Duffy[5], Brian Medeiros[5], and Matvey Debolskiy[2]

[1]Scripps Institution of Oceanography, University of California at San Diego, La Jolla, CA, USA
[2]Department of Geosciences, University of Oslo, Oslo, Norway
[3]Department of Meteorology, Stockholm University, 106 91 Stockholm, Sweden
[4]Research Institute for Applied Mechanics, Kyushu University, Fukuoka, Japan
[5]NSF National Center for Atmospheric Research, Boulder, CO, USA
[6]Pacific Northwest National Laboratory, Richland, WA, USA

**Correspondence:** Brandon M. Duran (bmduran@ucsd.edu)

**Abstract.**

Aerosol-cloud interactions (ACI) are a leading source of uncertainty in estimates of the historical effective radiative forcing (ERF) of climate change. One reason for this uncertainty is the difficulty of estimating the ERF from aerosol-cloud interactions (ERFaci) in climate models, which typically requires multiple calls to the radiation code and cannot disentangle the contributions from different processes to ERFaci. Here, we develop a new, computationally efficient method for estimating the shortwave (SW) ERFaci from liquid clouds using histograms of monthly-averaged cloud fraction partitioned by cloud droplet effective radius ($r_\mathrm{e}$) and liquid water path (LWP). Multiplying the histograms with SW cloud radiative kernels gives the total SW ERFaci from liquid clouds, which can be decomposed into contributions from the Twomey effect, LWP adjustments, and cloud-fraction (CF) adjustments. We test the method with data from five CMIP6-era models, using the Moderate Resolution Imaging Spectroradiometer (MODIS) satellite instrument simulator to generate the histograms. Our method gives similar total SW ERFaci estimates to other established methods in regions of prevalent liquid cloud, and indicates that the Twomey effect, LWP adjustments, and CF adjustments have contributed -0.34 ± 0.23, -0.22 ± 0.13, and -0.09 ± 0.11 Wm⁻², respectively, to the effective radiative forcing of the climate since 1850 in the ensemble mean (95 % confidence). These results demonstrate that widespread adoption of a MODIS $r_\mathrm{e}$– LWP joint histogram diagnostic would allow the SW ERFaci and its components to be quickly and accurately diagnosed from climate model outputs, a crucial step for reducing uncertainty in the historical ERF.

## 1 Introduction

Uncertainty in the historical effective radiative forcing from aerosol-cloud interactions (ERFaci) has remained stubbornly large across multiple generations of community assessments. The most recent Intergovernmental Panel on Climate Change (IPCC) assessment report estimated ERFaci between 1750 and 2019 to be -0.84 Wm⁻² ± 0.61 Wm⁻², compared to an estimate of +2.16 ± 0.26 Wm⁻² for the $CO_2$ radiative forcing over this period (90 % confidence) (Forster et al., 2021). Better constraints on the historical ERFaci are crucial for projecting future warming under different emission scenarios (Bellouin et al., 2020; Watson-





Parris and Smith, 2022; Wang et al., 2021), as well as for estimating the current level of committed anthropogenic warming (Mauritsen and Pincus, 2017; Dvorak et al., 2022). The large uncertainty of ERFaci means that it is equally likely that either (a) aerosol effects have masked the majority of historical $CO_2$-induced warming and future warming will accelerate as aerosol

emissions decrease (Andreae et al., 2005), or (b) ERFaci has played a small role in Earth's radiation balance over the historical period which implies relatively weaker future warming.

Reducing uncertainty in the historical ERFaci requires us to better understand the different processes that contribute to ERFaci. Aerosols affect liquid clouds by serving as a source of additional cloud condensation nuclei (CCN), leading to an increase in cloud droplet number concentration ($N_d$) and a decrease in cloud droplet effective radius ($r_e$) for a given cloud

liquid water path (LWP). This effect is known as the Twomey effect (Twomey, 1977), and it enhances cloud albedo because distributing a fixed amount of cloud liquid across a larger number of smaller droplets increases the total surface area of the droplets. The Twomey effect, or instantaneous radiative forcing (IRFaci), cools the climate system by increasing the amount of reflected shortwave (SW) radiation, and is distinguished from the macrophysical cloud adjustments which occur on longer timescales of hours in response to the elevated $N_d$/reduced $r_e$ cloud state (Gryspeerdt et al., 2021). These adjustments are

commonly separated into the LWP and cloud fraction (CF) adjustments, representing changes in cloud thickness and cloud amount, respectively.

Both the LWP and CF adjustments are the net effect of competing processes which can act to either increase or decrease the ERFaci, with precipitation suppression (Albrecht, 1989) and enhanced evaporation and entrainment (Ackerman et al., 2004; Wang et al., 2003) hypothesized to be the relevant adjustment mechanisms. A shift to smaller cloud droplets results in slower

droplet fall speeds and/or less droplet coalescence and growth into precipitation, suppressing precipitation and prolonging the lifetime of clouds. This lifetime effect can lead to an increase in cloud fraction and cloud LWP, hence higher cloud albedo, amplifying the forcing associated with the Twomey effect. Separately, decreased sedimentation fluxes and stronger evaporation rates in response to reduced droplet size tend to diminish cloud LWP (Jiang et al., 2006; Xue and Feingold, 2006; Bretherton et al., 2007). The timescale of these mechanisms exceed that of the Twomey effect, taking up to tens of hours to produce

changes in LWP (Glassmeier et al., 2021; Gryspeerdt et al., 2021).

Many estimates of the Twomey effect and the cloud adjustments have been made in observational and global climate model (GCM) studies, but the uncertainties in these components of ERFaci remain large. The latest World Climate Research Programme assessment gives 90% confidence ranges of -1.46 to -0.22 Wm$^{-2}$ for the Twomey effect, -0.06 to 0.88 Wm$^{-2}$ for the LWP adjustment, and -1.88 to 0.16 Wm$^{-2}$ for the CF adjustment (Bellouin et al., 2020), while the Sixth Assessment Report

(AR6) of the IPCC gave similar, but slightly narrower estimates. These estimates show that all three components of the SW ERFaci contribute substantial uncertainty, though the sum of the three terms' uncertainty ranges is much larger than the range of the total SW ERFaci, indicating correlation between the terms. Narrowing uncertainty in the total ERFaci requires improved understanding of the Twomey effect and the cloud adjustments, as well as the relationships between them.

Despite being investigated often, a key reason for the uncertainties in the components of ERFaci is the difficulty of quan-

tifying them in GCMs and observations. Several methods have been developed for estimating ERFaci in GCMs, but these are either prohibitively expensive or cannot decompose the total ERFaci into its components. The Ghan (2013) method per-



forms additional calls to the radiation code that neglect absorption and scattering by aerosols and clouds, so that comparing the aerosol-free and aerosol- and cloud-free output allows the ERFaci to be directly estimated. However, by making multiple calls to the radiation code and requiring a separate set of aerosol and cloud-free diagnostics that are not included in standard

model output, this method is difficult to implement in a large ensemble of models. A more accessible method for computing ERFaci is the approximate partial radiative perturbation (APRP) proposed by Taylor et al. (2007) as a modification to the partial radiative perturbation (PRP) method (Colman and McAvaney, 1997; Wetherald and Manabe, 1988). PRP relies on high-frequency (3-hourly) output and also makes extra calls to the radiation scheme (Mülmenstädt et al., 2019), so APRP has become a widely-adopted alternative thanks to (i) its reliance on standard model output at monthly frequency, (ii) the lack of

aerosol-free diagnostics required, and (iii) relatively strong agreement with the more accurate, but expensive, PRP technique (Taylor et al., 2007). APRP further distinguishes itself from the Ghan (2013) method because it can decompose the aerosol direct effect into absorption and scattering components, and the aerosol indirect effect into absorption, scattering, and amount components (Zelinka et al., 2023). However, it has been shown to be biased relative to more exact techniques due to some of its assumptions, and it cannot separately determine the contributions from the Twomey effect and the LWP adjustment (Zelinka

et al., 2014). Under APRP, these distinct effects are combined into a cloud scattering component, prohibiting an assessment of their relative contributions to the total SW ERFaci (Smith et al., 2020), as well as preventing a systematic comparison of these estimates to those derived from other methods. The ISCCP cloud radiative kernel (CRK) method shares similar strengths and limitations to APRP, but relies on output from the ISCCP satellite simulator (Klein and Jakob, 1999; Webb et al., 2001). Underlying all of these methods is an inability to decompose ERFaci into contributions from the Twomey effect, LWP adjustment,

and CF adjustment, which prevents an assessment of their relative contributions to the historical ERF.

There is thus a clear need for an inexpensive method of computing SW ERFaci in GCMs that can decompose the total aerosol forcing into contributions from the three cloud processes. Here, we present a new technique for computing SW ERFaci from liquid clouds in GCMs that addresses these issues with existing techniques. We introduce a MODIS-based cloud radiative kernel method in climate models that extends the observational-based approach from Wall et al. (2023) and can be used to

diagnose ACI and cloud feedbacks. Our new method relies on monthly joint histograms of cloud fraction generated by the MODIS satellite instrument simulator, matching APRP in its need for monthly output. Moreover, using a modified version of the cloud kernel technique pioneered in the cloud-feedback literature (Zelinka et al., 2013), our method is capable of separately calculating the radiative effects induced by the Twomey effect, LWP, and CF adjustments, allowing us to estimate their relative contributions to the total SW ERFaci from liquid clouds.

In what follows, we first describe the MODIS joint histograms of cloud fraction that form the core of our method, outline the steps for calculating the associated cloud radiative kernels and describe the decomposition of SW ERFaci using our MODIS cloud radiative kernel (MODIS CRK) method in Sect. 2. In Sect. 3, we apply the method to an ensemble of five CMIP6-era models and compare our results to existing approaches for estimating SW ERFaci. Next, in Sect. 4 we leverage a set of nudged historical simulations to evaluate how accurately GCMs simulate liquid-cloud fraction compared to observations, and discuss

how mean state biases in models imprint onto our estimates of SW ERFaci. Direct comparison with MODIS observations





enables us to construct potential emergent constraints (Williamson et al., 2021) on both the Twomey effect and the LWP adjustment and highlights insights that might be found with a larger ensemble. We finish with conclusions in Sec. 5.

## 2   Data and Methods

### 2.1   GCM simulations

Data are taken from simulations with four CMIP6-generation GCMs: the Energy Exascale Earth System Model v. 2 (E3SMv2, Golaz et al., 2022), the Norwegian Earth System Model v. 2 (NorESM2, Seland et al., 2020), the Model for Interdisciplinary Research on Climate v. 6 (MIROC6, Tatebe et al., 2019), and the Community Earth System Model v. 2 (CESM2, Danabasoglu et al., 2020). Two distinct sets of simulations were performed with MIROC6: one with the standard diagnostic precipitation scheme and a second with a prognostic precipitation scheme described in Michibata et al. (2019). These will be labeled as

MIROC6-DP and MIROC6-PP, respectively. E3SMv2 and NorESM2 use cloud schemes that have been modified from their CMIP6 counterparts (Golaz et al., 2022; Debolskiy, 2024; Olivié, 2024).

Our analysis uses pairs of atmosphere-only simulations with the same prescribed climatological annual cycles of sea-surface temperature and sea-ice fraction, but different aerosol concentrations. Atmospheric $CO_2$ concentrations are prescribed at preindustrial levels (284 ppmv) and the sea-surface temperature and sea ice concentrations are obtained by calculating the average

annual cycles from the final 100 years of a 4600-year fully-coupled preindustrial control simulation with the Geophysical Fluid Dynamics Laboratory CM3 GCM (Donner et al., 2011; Griffies et al., 2011). Using this lower boundary condition, we run one simulation with each model with preindustrial (1850) aerosol emissions and one with present-day (2000) aerosol emissions. We refer to these as CTL and PDaer, respectively. Each simulation is run for 10 years, with the first year discarded as spin-up and monthly-averaged output saved. All model data are regridded to a common $1° \times 1°$ resolution to match the equal-angle

grid of MODIS observations (King et al., 2003). We also make use of nudged historical simulations (HIST) over part of the MODIS observational period; the details of the nudging procedures employed for the simulations differ across the GCMs and are summarized in Table A1.

We analyze joint histograms that represent liquid-topped clouds partitioned by $r_e$ and LWP (Fig. 1a). The histograms are produced by the MODIS satellite instrument simulator, which is part of the COSP satellite simulator package (Bodas-Salcedo

et al., 2011; Swales et al., 2018) and emulates what the satellite would observe if it were orbiting above the model atmosphere. $r_e$ is estimated using a simplified pseudoinversion that relies on two lookup tables, one for liquid clouds and one for ice clouds, each summarizing the optical properties of cloud particle size distributions as a function of effective radius (for details, see Pincus et al. (2012)). LWP is estimated from $r_e$ and optical depth $\tau$ at each pixel. The $r_e$–LWP joint histograms produced by the MODIS simulator are intended to mirror their observational counterparts, detailed in Pincus et al. (2023). Recently, Wall

et al. (2023) used the observed joint histograms to estimate aerosol indirect effects from marine liquid clouds. The present work features the first uses of the new MODIS $r_e$–LWP joint histograms in GCMs and we hope to motivate more widespread adoption of this diagnostic.





The MODIS joint histograms are similar to the broadly utilized histograms from the International Satellite Cloud Climatology Project (ISCCP) (Webb et al., 2001), but with the added benefit that they can be further partitioned by cloud-top phase.

Moreover, the MODIS histograms have dimensions of $r_e$ and LWP, whereas the ISCCP histograms have dimensions of cloud-top pressure and optical depth. Phase discrimination is derived from several tests in the visible, near-infrared, and infrared portions of the spectrum (King et al., 2010), and is based on cloud condensate between the top of the highest cloud and one optical depth unit below. When visible extinction over this interval is at least 70 % from liquid droplets, the cloud is classified as liquid, and vice-versa. When over 70 % of visible extinction over this interval cannot be attributed to a single phase the

cloud is classified as "undetermined" phase. This label includes both true mixed-phase clouds and cases where an ice cloud with optical thickness between 0.3 and 0.7 occurs above a liquid cloud. We focus here on joint histograms of liquid-topped clouds.

## 2.2 Shortwave Cloud Radiative Kernel

We compute SW cloud radiative kernels for the MODIS $r_e$–LWP joint histograms using the RRTMG radiative transfer model

(Clough et al., 2005). The kernels represent the SW radiative flux anomaly at the top of the atmosphere that would occur given a unit increase in liquid-cloud fraction in a given histogram bin, holding all non-cloud factors fixed to their average annual cycle. The $r_e$ and LWP bin edges are 4.0, 8.0, 10.0, 12.5, 15.0, 20.0, and 30.0 μm and 0, 10, 30, 60, 100, 150, 250, and 20000 gm$^{-2}$, respectively.

Let $R$ represent the SW radiative flux at the top of the atmosphere and let $C_{pl}$ represent the liquid-cloud fraction in $r_e$ bin

$p$ and LWP bin $l$. For a specific latitude, surface albedo, and calendar month, the kernel $\mathbf{K}$ measures how anomalies of $C_{pl}$ change $R$ while holding all non-cloud factors fixed:

$$\mathbf{K} \equiv \frac{\partial R}{\partial C_{pl}} \tag{1}$$

The kernel methodology is derived from Zelinka et al. (2012a) with minor adjustments, following the procedure described in Wall et al. (2023). Here we highlight key differences that distinguish our approach from both of the above methods. (i) *Cloud*

*phase and kernel dimensions*: our cloud radiative kernel solely represents liquid-topped clouds and is partitioned by $r_e$ and LWP, to align with the MODIS joint histograms. The kernels in Zelinka et al. (2012a) represent clouds of all phases and are partitioned by cloud optical thickness and cloud-top pressure. (ii) *RRTMG input:* inputs to the model are derived from the climatological seasonal cycles of humidity, temperature, and surface skin temperature derived from the CTL simulations. These variables are first zonally-averaged and then averaged across the five GCM configurations before being input to the radiative transfer

model. Wall et al. (2023) use temperature and humidity profiles from reanalysis data as inputs. (iii) *Kernel interpolation:* we utilize the climatological seasonal cycle of clear-sky surface albedo from the CTL simulation for each individual GCM to linearly interpolate the cloud radiative kernel $\mathbf{K}$ from latitude-surface-albedo space to latitude-longitude space (Fig. 1b). This produces a distinct transformed cloud radiative kernel for each individual model which is a function of calendar month, latitude, longitude, $r_e$ and LWP, and has units of Wm$^{-2}$%$^{-1}$ (watts per square meter per percentage of cloud fraction). In Wall





et al. (2023), the interpolation is performed using clear-sky surface albedo from observations. The cloud radiative kernel is available at https://zenodo.org/records/13839356 (Duran, 2024).

## 2.3 SW ERFaci Decomposition

Once the SW cloud radiative kernel for the MODIS $r_e$–LWP joint histogram is linearly interpolated into latitude-longitude space for each GCM, we modify the cloud-feedback framework developed by Zelinka et al. (2013) to decompose the SW radiative flux anomaly at the top of the atmosphere induced by changes in liquid-topped clouds between CTL and PDaer into contributions from different cloud properties. For a given latitude, longitude, and calendar month, the total liquid-cloud-induced SW radiative flux anomaly at the top of atmosphere, $R'$, is generated by multiplying the cloud radiative kernel **K** by the change in cloud fraction histogram $C'$ and summing over all $r_e$ and LWP bins:

$$R' = \sum_{p=1}^{6} \sum_{l=1}^{7} (K_{pl} C'_{pl}),$$ (2)

where $C'$ represents the change in liquid-cloud fraction between PDaer and CTL. $R'$ gives an estimate of the contribution of liquid-topped clouds to the change in top-of-atmosphere (TOA) radiation associated with a perturbation, in this case from aerosols. The calculation of $R'$ is performed separately for each calendar month and then averaged over the seasonal cycle.

We decompose the right-hand side of Eq. (2) to estimate how much $r_e$, LWP, and cloud-amount anomalies contribute to $R'$, closely following the methodology outlined in Appendix B of Zelinka et al. (2013), but with different terms in the decomposition due to the distinct dimensions of the MODIS joint histogram. First, we decompose the cloud fraction anomaly into two terms. Let $C_{\text{tot}}$ represent the total liquid-cloud fraction summed over all histogram bins:

$$C_{\text{tot}} = \sum_{p=1}^{6} \sum_{l=1}^{7} C_{pl}.$$ (3)

We express the cloud fraction anomaly as

$$C'_{pl} = \frac{\overline{C}_{pl}}{\overline{C}_{\text{tot}}} C'_{\text{tot}} + C^*_{pl},$$ (4)

where overbars indicate values from the local climatological seasonal cycle. The first term on the right-hand side of Eq. (4) represents the contribution to $C'_{pl}$ from a change in cloud cover if $C'_{\text{tot}}$ were distributed across the $r_e$–LWP bins such that the normalized cloud distribution in the joint histogram space remains the same as climatology. Put differently, this term accounts for the change in total liquid-cloud fraction, proportioned according to the climatological distribution. The second term on the right-hand side captures any anomalies of $C_{pl}$ that remain after removing $(\overline{C}_{pl}/\overline{C}_{\text{tot}})C'_{\text{tot}}$; i.e., shifts in the cloud distribution across $r_e$ and LWP, holding total liquid-cloud fraction fixed. $C^*_{pl}$ vanishes when it is summed over all $r_e$–LWP bins by design.

Next, we decompose the radiative kernel into two terms:

$$K_{pl} = K_0 + \hat{K}_{pl}.$$ (5)





$K_0$ is the average of $K_{pl}$ weighted by the climatological cloud fraction,

$$K_0 \equiv \sum_{p=1}^{6} \sum_{l=1}^{7} \left( \frac{\overline{C_{pl}}}{\overline{C}_{\text{tot}}} K_{pl} \right), \tag{6}$$

and $\hat{K}_{pl} \equiv K_{pl} - K_0$. Combining the relationships from Eqs. (2)-(6), the total liquid-cloud-induced SW flux anomaly at TOA is given by

$$R^{'} \equiv \sum_{p=1}^{6} \sum_{l=1}^{7} (K_{pl} C_{pl}^{'}) = K_0 C_{\text{tot}}^{'} + \sum_{p=1}^{6} \sum_{l=1}^{7} (\hat{K}_{pl} C_{pl}^{*}), \tag{7}$$

where $K_0 C_{\text{tot}}^{'}$ represents the liquid-cloud-induced SW flux anomaly at TOA that would occur given a change in total liquid-cloud fraction, distributed across the climatological distribution of clouds in the $r_{\text{e}}$–LWP joint-histogram space. This term is

the cloud fraction adjustment, or the "proportionate change in cloud fraction" described in Zelinka et al. (2012b).

Next, we further decompose $\hat{K}_{pl}$ into three terms:

$$\hat{K}_{pl} = \hat{K}_p + \hat{K}_l + \hat{K}_{\text{res}}, \tag{8}$$

where

$$\hat{K}_p = \sum_{l=1}^{7} \left( \hat{K}_{pl} \sum_{p=1}^{6} \frac{C_{pl}}{C_{\text{tot}}} \right), \tag{9}$$


$$\hat{K}_l = \sum_{p=1}^{6} \left( \hat{K}_{pl} \sum_{l=1}^{7} \frac{C_{pl}}{C_{\text{tot}}} \right), \tag{10}$$

and

$$\hat{K}_{\text{res}} = \hat{K}_{pl} - \hat{K}_p - \hat{K}_l. \tag{11}$$

We can then express $R^{'}$ as

$$R^{'} = K_0 C_{\text{tot}}^{'} + \sum_{p=1}^{6} \left( \hat{K}_p \sum_{l=1}^{7} C_{pl}^{*} \right) + \sum_{l=1}^{7} \left( \hat{K}_l \sum_{p=1}^{6} C_{pl}^{*} \right) + \sum_{p=1}^{6} \sum_{l=1}^{7} (\hat{K}_{\text{res}} C_{pl}^{*}), \tag{12}$$

or

$$R^{'} = R_{\text{CF}}^{'} + R_{r_{\text{e}}}^{'} + R_{\text{LWP}}^{'} + R_{\text{res}}^{'}. \tag{13}$$

The first term on the right-hand side of Eqs. (12) and (13), $R_{\text{CF}}^{'}$ represents the contribution of changes in the total liquid-cloud fraction to the SW flux anomaly $R^{'}$. The second term is generated by multiplying an effective kernel that accounts for variations

in $r_{\text{e}}$ by the change in cloud fraction at each $r_{\text{e}}$ bin, and represents the contribution of $r_{\text{e}}$ anomalies to the SW flux anomaly $R^{'}$ with LWP and total liquid-cloud fraction held fixed. The third term on the right-hand side of Eqs. (12) and (13) is similar





to the second, but represents the contribution of LWP anomalies to $R'$ with $r_e$ and total liquid-cloud fraction held fixed. These terms are computed from the differences between PDaer and CTL, where SSTs are identical but aerosol emissions differ, hence they represent the CF adjustment, Twomey effect, and LWP adjustment, respectively. Each term is computed with the other properties held fixed, and the last term on the right-hand side ($R'_{\text{res}}$) is the residual of the decomposition. In this way, we are able to capture the distinct roles of changes in different cloud properties to the overall SW ERFaci.

The last step of the decomposition is to perform a correction to account for obscuration effects from non-liquid clouds. MODIS reports the cloud-top pressure and cloud phase of the highest cloud in each column. If an increase in non-liquid cloud occurs, MODIS may report a decrease in liquid-cloud fraction due to greater obscuration by non-liquid cloud, and vice-versa, even if there is no change in liquid cloud (see Fig. 1 in Zelinka et al., 2024). To control for obscuration effects, we perform a change of variable, replacing $C_{\text{tot}}$ with $[C_{\text{tot}}/(100\% - I_{\text{tot}})]'(100\% - \overline{I}_{\text{tot}})$, where $I_{\text{tot}}$ is the non-liquid cloud fraction reported by MODIS. Utilizing the non-liquid cloud fraction in conjunction with the total cloud fraction enables us to verify that the cloud fraction changes seen by the MODIS simulator are not contaminated by free-tropospheric cloud changes (Scott et al., 2020; Zelinka et al., 2024). Code to perform the SW ERFaci decomposition is available to download at the url linked in the Code and Data Availability Section.

## 3 Results

### 3.1 SW ERFaci diagnosed using the MODIS-cloud radiative kernel method

Here we present estimates of the total SW ERFaci, the Twomey effect and the cloud adjustments for the five GCMs using our new method. Figure 2a shows the ensemble-mean total SW ERFaci from liquid clouds; Fig. 2b-d show the three terms of the SW ERFaci decomposition; Table 1 contains the global-mean values for the total SW ERFaci and each of the three components for the individual models; and Figure A1 shows the MODIS CRK decomposition for the individual GCMs.

Our method gives an ensemble- and global-mean total SW ERFaci of -0.75 Wm$^{-2}$, close to the mean value in the latest IPCC assessment, though that estimate is for clouds of all phases and includes longwave (LW) ERFaci (Forster et al., 2021). There is large intermodel spread across the ensemble, but all models agree on a negative total SW ERFaci from liquid clouds (Fig. 2a, Table 1). CESM2 exhibits the most negative SW ERFaci, consistent with Zelinka et al. (2023), where it had the second-strongest SW ERFaci of all CMIP6 models assessed. The two versions of MIROC6 have the weakest SW ERFaci, because they have less liquid cloud in the control state, so fewer clouds susceptible to aerosol perturbations and weaker aerosol forcing. These systematic differences are discussed further in Sect. 3.2.3. The ensemble-mean aerosol forcing is hemispherically asymmetric (Fig. 2a), with Northern Hemisphere (NH) SW ERFaci nearly three times larger than Southern Hemisphere (SH) SW ERFaci. The strongest negative forcing is over highly industrialized regions and their outflows – East Asia, the Eastern United States and Europe – and in the stratocumulus regions off the coasts of western South America, Africa, and Australia.

The Twomey effect, or IRFaci, is negative in all models (Fig. 2b). Most of the NH is cooled by the IRFaci, with the strongest brightening again over and downstream of industrial regions in East Asia and parts of Europe and North America. The forcing is weaker in the SH, commensurate with the less aerosol compared to the NH. An exception is seen in the subtropical





stratocumulus regions off the western coasts of South America and Africa, where extensive low cloud cover and relatively clean background states result in large negative IRFaci despite relatively small changes in emissions between CTL and PDaer in these regions. The ensemble-mean spatial pattern of the Twomey effect broadly agrees with estimates from observational (Jia et al., 2021; McCoy et al., 2017; Wall et al., 2023) and GCM studies (Gryspeerdt et al., 2020; Mülmenstädt et al., 2019), though McCoy et al. (2017) and Gryspeerdt et al. (2020) report stronger cooling in the Northeast Pacific (NEP) stratocumulus

region. With the exception of land regions near anthropogenic aerosol sources, the MODIS CRK decomposition also captures the strong land-ocean forcing contrast found in observational estimates (McCoy et al., 2017; Diamond et al., 2020), as cleaner background conditions and high cloud droplet number ($N_d$) susceptibility to aerosol perturbation typically produce larger aerosol forcing over the ocean.

The LWP adjustment is generally negative, consistent with the tendency of GCMs to predict uniform LWP increases in

response to aerosol perturbations (Bellouin et al., 2020). The spatial pattern of the LWP adjustment is more consistent with the total SW ERFaci ($r = 0.72$ ensemble-mean pattern correlation between LWP adjustment and SW ERFaci) than that of the Twomey effect ($r = 0.57$), but the LWP response is generally weaker than the IRFaci. We note that GCMs typically suggest a positive LWP response but observational studies are more equivocal, with some finding a decrease in LWP in response to aerosol perturbations (Chen et al., 2014; Sato et al., 2018; Wall et al., 2022), others bi-directional LWP responses (Ackerman

et al., 2004; Michibata et al., 2016; Toll et al., 2019; Possner et al., 2020; Zhang et al., 2022) and still others see little change in LWP (Malavelle et al., 2017; Toll et al., 2017). The conflicting evidence between GCMs and observations may be related to the asymmetric representation of positive and negative LWP processes in GCMs. While evaporation and entrainment are both processes that can contribute to reduced cloudiness in response to smaller, more numerous cloud droplets, large-scale models have disparate representations of them, with the representation of cloud-top entrainment especially variable. By parameterizing

precipitation suppression but not enhanced evaporation and entrainment feedbacks, models may not capture the full range of physics that govern LWP responses to aerosol perturbations, hence producing incomplete estimates of the cloud adjustments (Zhou and Penner, 2017; Mülmenstädt et al., 2019). However, recent work from Mülmenstädt et al. (2024) suggests that even in models that exhibit entrainment-like behavior, negative LWP responses due to cloud-top entrainment enhancement may only play a small role in the global-mean LWP adjustment. This indicates that more refined representations of entrainment in GCMs

may not resolve the disagreement between GCMs and observations. We highlight this discrepancy to indicate that the results of our technique fall along the existing divide of GCM and observational results.

Finally, the CF adjustment is generally smaller than the other adjustments, though CESM and NorESM2 both have substantial global-mean adjustments of -0.233 and -0.157 Wm$^{-2}$, respectively. The spatial pattern of the CF adjustment is also more heterogeneous, with regions of both positive and negative forcing, and shows little agreement across the models. The

diversity of spatial structures across the ensemble may be due to differences in the mean state liquid- and ice-cloud fraction distributions in CTL, as well as differences in circulation changes, which will impact the changes in cloud fraction. Using MODIS observations, Gryspeerdt et al. (2016) found an enhanced CF adjustment at the edges of stratocumulus regions in observations, where they hypothesize aerosols play a dominant role in modulating the stratocumulus-to-cumulus transition (see also Gryspeerdt et al., 2020, who found a similar effect in model data), but we do not see evidence of such systematic behavior





in our decomposition. Instead, the relatively small, heterogeneous contribution of the CF adjustment to the total SW ERFaci
found here agrees with the analysis of CMIP5 and CMIP6 models by Zelinka et al. (2023).

## 3.2  Comparison to existing methods

### 3.2.1  Total SW ERFaci comparison

In this section, we seek to build confidence in our method by comparing with existing methods to diagnose SW ERFaci: APRP
and the ISCCP cloud radiative kernel technique. We do not compare with the double-call method of Ghan (2013) due to its
need for aerosol-free and aerosol- and cloud-free output; however, prior work (Zelinka et al., 2023) has shown that the APRP
method yields aerosol ERF values that agree closely, both in the global-mean and spatially, with the double call method. While
benchmarking our method is important to establish its validity, direct comparisons with existing methods are not possible
because our method only considers aerosol impacts on liquid-topped clouds, while other methods diagnose aerosol forcing on
clouds of all phases. We address this complication by focusing our comparison on regions dominated by liquid clouds, defined
as grid points with an annual-mean non-liquid cloud fraction of less than 10 %. The points sampled by this threshold can be
seen in Fig. A2 and include large swaths of the eastern subtropical ocean basins, as well as desert regions and sections of the
eastern tropical Pacific.

Figure 3 shows point-by-point comparisons over these regions of the total SW ERFaci from liquid clouds computed using
MODIS cloud radiative kernels and the total SW ERFaci from all clouds computed using the APRP (top row) and ISCCP
CRK (bottom row) methods[1]. Our estimates for CESM2 and NorESM2 show strong agreement with both methods, with $r^2$
values of 0.8 and higher, and slopes approaching 1. The agreements are weaker for E3SMv2 ($r^2$ of 0.57 and 0.54, respectively),
despite sharing similar non-liquid cloud fractions with NorESM2 and CESM2 in the validation regions. Finally, our estimates
for the two MIROC models are well correlated with the estimates from the other methods, but have large deviations from the
one-to-one line.

The deviations from the one-to-one line for MIROC6-DP (Fig. 3d, i) and MIROC6-PP (Fig. 3e, j) can be explained by the
small liquid-cloud fractions in these models. While we find that the other three GCMs are successful in identifying cloud phase
for more than 95 % of clouds, in MIROC6 the MODIS simulator fails to classify over 50 % of the clouds it observes into a
distinct thermodynamic phase (Fig. A4). So applying a threshold based on only non-liquid CF (which includes both ice and
undetermined clouds) constrains our validation in these models to the Sahara and parts of the North Atlantic (Fig. A2). Typical
regions where liquid clouds are prevalent, such as the stratocumulus decks, are not included because of the undetermined cloud
bias in these regions. The remaining regions, while characterized by low amounts of non-liquid cloud, also lack a substantial
amount of liquid clouds. Therefore, the majority of grid points evaluated in this comparison show a SW ERFaci from liquid
clouds of approximately zero. In contrast, total SW ERFaci computed using the APRP and ISCCP CRK methods includes
clouds of all phases, potentially giving large values of SW ERFaci. A more restrictive threshold (i.e. low non-liquid CF and

---

[1]Figure A3 demonstrates the strong point-by-point agreement between total SW ERFaci as calculated by the APRP and ISCCP CRK methods, confirming
that these existing methods have been implemented correctly.



high liquid CF) could serve to fine-tune our comparison; however, this would exclude both MIROC6 models by default and thus we do not pursue it further. We discuss other implications of the cloud-top phase identification bias in Sect. 4.1. Overall, our MODIS CRK method compares well with established methods in regions where liquid clouds are dominant, except in MIROC6 where it is strongly influenced by the cloud-top phase identification bias.

**3.2.2 Comparison of the three adjustments**

One of the principal advantages of our approach is the ability to separately diagnose the Twomey effect and cloud adjustments using only monthly output and no additional calls to the radiation code. APRP is widely used due to similar benefits (monthly output, only eight fields that are routinely diagnosed in models required), but combines the Twomey effect and LWP adjustment into one term. Smith et al. (2020) proposed a procedure for separating the APRP cloud-scattering term into these distinct

contributions based on the strong linear relationship between global-mean in-cloud LWP and the strength of the global-mean LWP adjustment found in Gryspeerdt et al. (2020), as models with the largest increase in in-cloud LWP between their respective CTL and PDaer simulations produced the strongest/most negative LWP adjustment. Then after estimating the LWP adjustment ($A_{\mathrm{LWP}}$), the Twomey effect (IRFaci) can be calculated as a residual of the APRP-calculated total SW ERFaci and the sum of the CF adjustment ($A_{\mathrm{AMT}}$) and $A_{\mathrm{LWP}}$:

$$\mathrm{IRFaci} = \mathrm{ERFaci} - A_{\mathrm{LWP}} - A_{\mathrm{AMT}}. \tag{14}$$

We implement this approximation in our APRP decomposition to compare with the Twomey effect and LWP adjustment diagnosed by our new MODIS CRK method, finding that the two methods give quite different results. While the LWP adjustment explicitly diagnosed by the MODIS CRK methodology is non-negligible and at least 25 % of the total SW ERFaci from liquid clouds across all five models, the APRP regression-estimated LWP adjustment is at most 10 % of the total SW ERFaci

(Table A2). As a result, since it is calculated as a residual in the Smith et al. (2020) APRP approximation, the Twomey effect appears as the dominant driver, ranging from 80-90 % of the APRP total SW ERFaci. Conversely, when diagnosed directly with our method, the Twomey effect is greater than the LWP adjustment in all cases, but not as dominant, ranging from 84 % of the total SW ERFaci in MIROC6-PP to only 35 % in CESM2. Not only is there disagreement in the global-mean quantities, but a point-by-point comparison of the two approaches indicates weak correlation between the two approaches for calculating

the LWP adjustment (not shown), regardless of whether we try to control for the amount of non-liquid clouds.

The strong disagreement with our method indicates drawbacks to relying on APRP for calculating the three components of SW ERFaci and suggests APRP may underestimate the LWP adjustment and overestimate the Twomey effect. One reason for this discrepancy may be that the estimate from Smith et al. (2020) uses total grid-box averaged cloud water path and cloud ice water path: this difference may include scenarios where LWP changes in deep clouds that contain ice at higher levels. It may

also be the case that LWP adjustment plays a larger role relative to the Twomey effect when considering liquid clouds only, but this is unlikely to fully explain the differences in the estimates. Under the Smith et al. (2020) approximation, errors and uncertainties in the LWP and CF adjustments will cause errors in the estimate of the Twomey effect, since it is calculated as a residual term. Considering that both the Twomey effect and LWP adjustment separately contribute substantial uncertainty to





estimates of SW ERFaci, we note these contrasting results to stress the importance of isolating each of the three components
of SW ERFaci and directly quantifying the Twomey effect rather than inferring it as a residual.

### 3.2.3 Impact of Prognostic Precipitation on ACI in MIROC6

The MIROC6 simulations enable us to qualitatively validate our decomposition with previous studies that have investigated
the impact of prognostic precipitation schemes on ERFaci (Michibata et al., 2019, 2020). Our ensemble features two versions
of MIROC, one with a diagnostic precipitation scheme that assumes that all diagnosed rainwater precipitates to the surface
within a single-model time step, and a second with a prognostic precipitation scheme in which precipitating hydrometeors can
stay suspended in the atmosphere across multiple model time steps. Previous studies (e.g., Gettelman et al., 2015; Michibata
et al., 2019) have documented how prognostic precipitation schemes can improve representation of the microphysics and
hydrometeor distribution due to enhanced accretion. Strengthening of the accretion-to-autoconversion ratio has been shown
to diminish the excessive cloud water susceptibility to aerosols found in diagnostic precipitation schemes and hence weaken
ACI, as the autoconversion rate is directly linked to ACI due to its dependence on $N_\mathrm{d}$ (Posselt and Lohmann, 2008, 2009). The
enhanced accretion and longer residence times of precipitation in a prognostic treatment lead to decreased $N_\mathrm{d}$ and strengthened
wet-scavenging of aerosols, with both effects decreasing cloud lifetimes (Michibata and Suzuki, 2020). The prognostic scheme
also influences cold-rain processes: due to the explicit representation of the riming process in MIROC6-PP, positive LWP
responses to aerosol perturbations can be further damped by falling snow originating higher in the atmosphere that remains in
the atmosphere across several time steps. This mechanism was dubbed the "snow-induced buffering" of ACI in Michibata et al.
(2020).

These effects of precipitation schemes on ACI can be interpreted through the lens of changes to $r_\mathrm{e}$ and LWP. With the
prognostic precipitation scheme, $r_\mathrm{e}$ decreases less between CTL and PDaer, and the LWP increase is also muted (not shown
here). As a result, MIROC6-PP has a nearly threefold reduction in the total SW ERFaci from liquid clouds (-0.33 Wm$^{-2}$ to
-0.128 Wm$^{-2}$) and in the LWP adjustment (-0.145 Wm$^{-2}$ to -0.046 Wm$^{-2}$) compared to MIROC6-DP (see also Fig. A5). The
SW ERFaci from liquid clouds weakens more when using a prognostic precipitation scheme than estimates in Gettelman et al.
(2015) and Michibata et al. (2019), but our results are supported by similarly strong reductions in total SW ERFaci of about
$\sim 50\,\%$, demonstrating that the larger ERFaci reductions in this study are not a reflection of our methodology, and instead are
a feature of the simulations. Spatially, the weakened ERFaci in MIROC6-PP is dominated by a large reduction over the NH
midlatitudes, where LWP responses to anthropogenic aerosols are strong, but susceptible to damping. We take the qualitative
agreement with prior studies as further evidence of the skill of our new method.





## 4    Understanding model spread in SW ERFaci components

### 4.1    Mean state biases in subtropical stratocumulus regions

The SW ERFaci decomposition from our new MODIS CRK method demonstrates substantial intermodel spread in the com-
ponents of SW ERFaci, especially the cloud adjustments. The structural differences between models complicate the process of
identifying causes of ensemble spread, but one source of this spread will be due to differences in initial states across the models.
To address this issue, we analyze nudged historical simulations that keep the large-scale circulation close to the observed state
of the atmosphere, reducing differences in model state and removing one source contributing to the ensemble spread in the
components of SW ERFaci.

We begin by comparing cloud distributions as functions of LWP and $r_e$ in the subtropical stratocumulus regions. We use
the same $20° \times 20°$ boxes as Zhang and Feingold (2023) to define the regions of five stratocumulus decks. These regions
are characterized by strong negative SW ERFaci, high susceptibility to aerosol perturbations and largely absent high cloud
cover. Therefore, they exert a disproportionate influence on the diagnosed SW ERFaci and mean state biases in these regions
could translate to large biases in our global-mean estimates of SW ERFaci from liquid clouds. Figure 4 shows liquid-cloud
fraction (LCF) histograms from MODIS observations (leftmost columns) and from the nudged historical simulations with the
five GCMs in the five stratocumulus regions. Note that we take averages over a common time period (2003–2014). Figure A6
shows the five stratocumulus regions analyzed.

In observations, stratocumulus regions are dominated by low LWP ($< 100$ gm$^{-2}$), small $r_e$ ($< 12.5$ µm) clouds, which we take
to be non-precipitating (left-most panels in Fig. 4). Each of the five regions varies in the amount of clouds at the opposite end of
the $r_e$–LWP spectrum (high LWP, large $r_e$), which we take to be precipitating. Zhang and Feingold (2023) also find that cloud
states of $r_e < 12.5$ µm and cloud states with LWP $< 50$ gm$^{-2}$ are persistent across the five regions in observations, while denser,
larger radii cloud states vary between basins. The Northeast Pacific (NEP), Southeast Pacific (SEP), and Australian (AUS)
cloud decks feature relatively high amounts of high LWP, large $r_e$ clouds; in contrast, the Northeast and Southeast Atlantic
(NATL, SEA) regions are more strongly skewed in the $r_e$–LWP space, with abundant non-precipitating clouds. The NEP, SEP,
and SEA regions are characterized by large total liquid cloud fractions ($> 50$ %), whereas cloud fractions are lower in the AUS
and NATL regions.

The models exhibit a large spread in their ability to reproduce these observed liquid cloud fractions and distributions. We
normalize LCF by the $r_e$–LWP bin with the largest liquid cloud fraction in each stratocumulus region and in each model (Fig.
4) in order to compare the distribution of cloud fraction in the $r_e$–LWP phase space separately from cloud fraction biases.
E3SMv2 most closely captures the amount of LCF simulated by MODIS and the cloud distribution in $r_e$–LWP space, in some
instances even simulating too many liquid clouds in the stratocumulus regions, whereas most models simulate too few clouds.
E3SMv2 also tends to most accurately simulate the predominance of low LWP, small $r_e$ clouds seen in observations. NorESM2
and CESM2 perform similarly to each other, consistently capturing around 80 % of the total observed MODIS LCF, though
they tend to be more skewed towards the high LWP, large $r_e$ regime. They exhibit similar cloud fractions to each other, with
the exception of the SEP for which NorESM2 simulates a substantially larger LCF. Finally, the two versions of MIROC6 have



large LCF deficits compared to observations and highly-skewed $r_e$–LWP cloud distributions, simulating a negligible amount of clouds with $r_e < 15$ μm. This may reflect an inability to simulate a sufficient amount of non-precipitating clouds, but we do not have direct evidence of precipitation state to confirm this. MIROC6 is known to suffer from the "too few, too bright" cloud problem (Nam et al., 2012), even when switching to a prognostic precipitation scheme (Michibata et al., 2019).

Mean state biases in the distribution of clouds in $r_e$– LWP space appear to be related to the SW ERFaci from liquid clouds. The three GCMs analyzed here that simulate the most negative SW ERFaci (E3SMv2, CESM2, NorESM2) accurately reproduce observed liquid cloud fractions and distributions, suggesting that estimates of aerosol forcing from these models may be more realistic. Conversely, MIROC6-DP and MIROC6-PP simulate weak SW ERFaci and have much too little LCF in their mean states. Their cloud fraction distributions are also strongly biased compared to observations, though it is difficult to assess

whether these distributional biases are linked to a bias in MODIS cloud-top phase classification (i.e., clouds that may be classified as liquid by MODIS, and therefore would be accounted for in our decomposition, are instead identified as "undetermined" phase) or reflect more fundamental issues with how the model simulates cloud coverage in stratocumulus regions.

## 4.2    Potential emergent constraints on components of SW ERFaci

The relationship between mean state biases and the historical SW ERFaci identified in the previous section suggest the potential

to use our MODIS CRK method to develop emergent constraints on the total SW ERFaci and its components. Here we present an example of how our method, paired with corresponding nudged historical simulations, might be used to form emergent constraints on SW ERFaci. We hypothesize that mean state liquid-cloud fraction serves as a first-order control on the strength of SW ERFaci, as greater amounts of simulated LCF implies the presence of more liquid clouds susceptible to aerosol perturbations and more negative SW ERFaci (Zhao et al., 2024). Given the small number of models in our ensemble, it is difficult

to establish robust emergent constraints; nevertheless, we assess whether this hypothesis holds true in our ensemble, as well as the ability of each model to reproduce the global-mean LCF retrieved from MODIS observations. We do this by comparing the mean states of the models in the historical simulations with the SW ERFaci diagnosed from the CTL and PDaer simulations. Although a true emergent constraint would require comparing the SW ERFaci from historical runs, the LCF characteristics within models are well-correlated between the CTL and nudged historical simulations (not shown).

We find strong relationships between mean state LCF in HIST and two of the SW ERFaci components: the global-mean Twomey effect and LWP adjustment (Fig. 5). Across our five models, the Twomey effect becomes increasingly negative as the mean LCF in HIST increases, ranging from -0.013 Wm$^{-2}$ for MIROC6-PP (19.3 % LCF in HIST) to -0.61 Wm$^{-2}$ for E3SMv2 (27.2 %). The same is largely true for the LWP adjustment, except that CESM2 simulates a slightly stronger adjustment than E3SMv2 ($\sim$ 0.01 Wm$^{-2}$) despite having slightly less LCF in the base state. This confirms that our hypothesis holds for the

models analyzed, though we note that much of the model spread comes from the outlier MIROC6 models.

     MODIS observations can be used in conjunction with the above relationships to give estimates of the Twomey effect and LWP adjustment. From 2003–2014, the global-mean LCF reported by the MODIS satellite is 27.1 $\pm$ 1.2 % (1$\sigma$ range), which is very close to the base state LCF of E3SMv2 and CESM2 and indicates that the strength of the aerosol forcing in these models





is more realistic. Using our potential emergent constraints, we estimate the historical Twomey effect and LWP adjustment to
be -0.55 ± 0.21 and -0.34 ± 0.13 Wm$^{-2}$, respectively (95 % confidence interval, CI).

These results suggest that mean state LCF could act as a powerful emergent constraint on both the Twomey effect and
the LWP adjustment. However, there are two caveats to the analysis presented here. First, a larger ensemble is required to
establish the robustness of the relationships identified in our analysis. Our analysis here should serve as an example as to how
the new MODIS CRK method can be leveraged to derive robust emergent constraints with a larger ensemble. Fortunately, the
low computational expense of our method, coupled with its ease of implementation and reliance on pre-existing fixed-SST
simulations, means that it should be possible to use larger ensembles in the near future. We also note that Fig. 6 demonstrates
that the models in our ensemble lie towards the higher end of the CMIP6 range for SW ERFaci (compare ensemble means of
Fig. 6b and d). Direct comparison of our estimates with those in Zelinka et al. (2023) is complicated by differing definitions of
present-day aerosols in the PDaer simulations (2000 for the simulations used here; 2014 for CMIP6 models in Zelinka et al.,
2023) and different prescribed SSTs and sea-ice concentrations, but it would be interesting to include CMIP6 models with
weaker total SW ERFaci estimates to test the robustness of our finding.

Second, the regression for the Twomey effect predicts a positive forcing for models with LCF < 17 % even though our
understanding of the Twomey effect says that it should still produce a cooling effect for small LCF. However, the slope of
the regression line for the Twomey effect is strongly influenced by the two MIROC6 models, which are also the furthest from
reality. So a larger ensemble might reveal that MIROC6-DP and MIROC6-PP are biasing the regression in an unphysical
direction.

These results are promising, and are supported by a strong physical rationale: more mean state LCF should be associated
with a stronger/more negative Twomey effect and LWP adjustment since there is more cloud for aerosols to perturb. However,
while our rationale for more mean state LCF being associated with a stronger/more negative LWP adjustment is consistent
with the GCM results, observational studies suggest there may be disagreement between GCMs and observations over the sign
of the LWP adjustment (see Introduction). As such, any emergent constraint on LWP should be interpreted with caution until
the causes of the model-observation discrepancy have been identified.

## 5   Conclusions

We have presented a new method of diagnosing the liquid SW ERFaci in models that output simulated MODIS cloud fractions
as functions of cloud droplet effective radius and cloud water path, partitioned by cloud-top phase. This method allows us to
efficiently estimate the total liquid SW ERFaci from monthly data, and to separately quantify the contributions of the Twomey
effect, the LWP adjustment and the CF adjustments to the total forcing. We tested the method using sets of simulations with
five models, including two variants of MIROC6, one with diagnostic precipitation and one with prognostic precipitation. Our
ensemble-mean estimate of liquid SW ERFaci (-0.75 Wm$^{-2}$) is close to the central total ERFaci estimate (-1.0 Wm$^{-2}$) from
the IPCC AR6 that includes a small positive offsetting effect of LW ERFaci (Forster et al., 2021). Our decomposition results
indicate that the Twomey effect (-0.34 Wm$^{-2}$) and the LWP adjustment (-0.22 Wm$^{-2}$) produce the majority of the cooling





ERFaci. While the CF adjustment is the smallest of the three components of SW ERFaci for each GCM, two models simulate substantial cooling (CESM2 and NorESM2, with -0.233 and -0.157 Wm⁻², respectively).

Our estimates for the components of SW ERFaci differ from the observational estimates in Wall et al. (2023), despite relying on the same MODIS CRK method. The GCM estimates of the Twomey effect (-0.34 ± 0.24 Wm⁻²) and cloud adjustments (-0.32 ± 0.29 Wm⁻²) averaged over ocean between 55°S and 55°N are only 44 % and 31 % as strong as the observational estimates from Wall et al. (2023) (-0.77 ± 0.25 and -1.02 ± 0.43 Wm⁻², respectively), but both approaches are similar in diagnosing comparable contributions from IRFaci and the cloud adjustments to the overall SW ERFaci (95 % confidence intervals). The weaker cloud adjustments in this study also obscure disagreement over the respective contributions of the LWP and CF adjustments: while Wall et al. (2023) find a significantly negative CF adjustment and near-zero or positive LWP adjustment from observations, we find opposing results more in line with prior GCM results (see Fig. 5 in Wall et al., 2023). A strong cooling effect from the Twomey effect and LWP adjustment, as well as a near-zero contribution to SW ERFaci from the CF adjustment, is a consistent feature of CMIP5 and CMIP6 models (Zelinka et al., 2023), suggesting that our method captures this broad characteristic of GCMs whilst also offering insight into the connections between the components of SW ERFaci to various physical processes and their representations in climate models.

We have compared our ERFaci estimates to two existing methods: the widely used APRP method and another method relying on the same Cloud Radiative Kernel approach used here (Zelinka et al., 2012a) but with output from the ISCCP satellite simulator that cannot partition by cloud-top phase and does not separate out cloud droplet effective radius and cloud water path. In liquid cloud-dominated regions where we would expect the total diagnosed forcing to be similar, the models show strong agreement on a point-by-point basis, though the agreement is somewhat worse for the E3SMv2 model. Neither existing method is able to separate out the three components of the total SW ERFaci, but Smith et al. (2020) proposed an approximation for the LWP adjustment in the APRP method. Our results disagree with this approximation, as we find a larger, and more realistic, contribution of the LWP adjustment to historical SW ERFaci.

We examine mean state biases within the models to investigate whether they help to explain the substantial ensemble spread in the components of SW ERFaci. Using a set of nudged historical runs, we find that in the subtropical stratocumulus regions, E3SMv2 simulates liquid-cloud fraction and distribution in $r_e$–LWP variable space that are very comparable to MODIS observations. CESM2 and NorESM2 perform similarly to each other, lacking the low LWP/small $r_e$ liquid cloud maxima seen in MODIS, but capturing close to 80 % of observed LCF. Both versions of MIROC6 simulate insufficient liquid clouds and negligible cloud cover with $r_e < 15$ μm. The effective radius bias is stronger with the prognostic precipitation scheme, and we also note that both variants of MIROC6 may overestimate the frequency of mixed-phase clouds. The cloud-top phase identification bias found in MIROC6 occurs uniformly in space. Whereas the other three GCMs feature enhanced levels of "undetermined" phase classification in areas where mixed-phase clouds are prevalent, including the NH extratropical storm track and the Southern Ocean (Zhang et al., 2010), no such spatial variability is found in MIROC6: areas like the stratocumulus regions, where liquid clouds should be dominant and ice clouds rare, show very similar biases compared to mixed-phase cloud-rich regions. The new MODIS $r_e$–LWP joint histogram diagnostic enables novel opportunities for model evaluation that can inform our interpretation of ERFaci estimates from GCMs.





Finally, we find that models with greater mean-state LCF simulate more negative SW ERFaci, which we expand upon further to investigate potential emergent constraints between mean-state LCF and the Twomey effect and LWP adjustment. Using these constraints in conjunction with MODIS observations, we obtain estimates of the historical Twomey effect (-0.55 $\pm$ 0.21 Wm$^{-2}$)

and LWP adjustment (-0.34 $\pm$ 0.13 Wm$^{-2}$) (95 % CIs). Models with global-mean liquid cloud fractions closest to MODIS observations (E3SMv2, CESM2) simulate the most cooling from the Twomey effect (-0.612 and -0.426 Wm$^{-2}$, respectively) and provide estimates that approach the magnitude of the observational estimate (-0.77 Wm$^{-2}$) from Wall et al. (2023). The low liquid cloud fractions in both MIROC6 versions influence the historical Twomey effect and LWP adjustment estimates, and suggest that GCMs that simulate insufficient amounts of liquid clouds will tend to underestimate the magnitude of the Twomey

effect and LWP adjustment. The constraint on the LWP adjustment disagrees in sign with observational estimates though, which undercuts its robustness and should be considered cautiously until the sources of the model-observation disparity have been discovered.

These results demonstrate the promise of our new technique for diagnosing SW ERFaci quickly and efficiently in GCMs, though our analysis is limited by the small size of the model ensemble. Furthermore, while our approach produces good

agreement with two existing methods for computing SW ERFaci, the comparison is narrow in scope because we calculate aerosol forcing on liquid clouds only. A more like-for-like comparison that can be applied globally to benchmark against other methods would build confidence in our approach. Wall et al. (2024) shows that the new MODIS CRK method, computed in cloud-top pressure and cloud optical depth space, agrees well with APRP when calculating cloud feedbacks, providing additional support to the comparison conducted here.

Despite these limitations, this work showcases the benefits of our new MODIS CRK method, which relies on monthly output from a pair of fixed-SST simulations that are standard parts of the RFMIP experimental protocol (piClim-control and piClim-aer in CMIP6; Pincus et al., 2016). Additionally, the code required for the new MODIS $r_e$–LWP joint histograms is already implemented in the latest version of COSP (CFMIP). The low computational expense and storage demands required by our approach should facilitate the generation of a larger model ensemble that could be used to test the validity of the emergent

constraints highlighted here.

The MODIS CRK method could also be adapted to investigate aerosol forcing from ice clouds, as well as to better understand the mechanisms driving ERFaci in GCMs. While the MODIS ice-cloud fraction histograms were not included in our simulations, the code required to output them as a diagnostic is included in the most recent COSP version (CFMIP). Future work implementing the MODIS CRK approach could be done with GCMs that represent aerosol interactions with ice clouds, which

would allow a more comprehensive validation of our estimates of SW ERFaci. Our method could also be used in a targeted analysis of a single GCM to yield deeper insights into the individual components of SW ERFaci. For example, a perturbed physics ensemble (PPE) could be used to examine how parameter uncertainty imprints onto each component of SW ERFaci, similar to Duffy et al. (2024), who used a CAM6 PPE to explore the core model processes controlling the large spread in cloud feedbacks. Similarly, Song et al. (2024) employed a PPE to investigate the buffering of ERFaci as a result of the interaction

between precipitation efficiency and radiative susceptibility. Implementing our new method in a PPE would enable each of the



components of SW ERFaci to be quantified and studied separately, and would provide another test of the emergent constraints proposed in Sect. 4.2.

The global constraints found here motivated us to search for similar constrains on a regional scale, but we have been unable to find any robust constraint across the five stratocumulus regions. One reason for this may be that at a local scale, large aerosol perturbations can generate nonlinear effects that are masked when averaging over the globe (Bellouin et al., 2020). Nevertheless, the new MODIS joint histogram diagnostic provides the ability to assess regional mean-state biases in liquid clouds across different models. This could be used to explore differences in model responses to local forcings; for instance, to evaluate how liquid clouds respond to aerosol perturbations under marine cloud brightening (MCB) scenarios in GCMs.

Our findings contribute to the ongoing work seeking to better constrain ERFaci. We provide what we believe are the first distinct estimates for the Twomey effect and cloud adjustments for an ensemble of CMIP6 models, which suggests that the LWP adjustment may contribute a comparable forcing to the Twomey effect. Our method also shows that differences in mean state liquid-cloud fraction drives differences in the components of SW ERFaci across models, indicating that some of the spread in SW ERFaci estimates may be reduced by improving representation of liquid clouds in global climate models. Applied within a larger ensemble, our technique offers promise for narrowing the significant uncertainty in our estimates of SW ERFaci and its components, helping improve our understanding of the historical effective radiative forcing.

*Code and data availability.* The SW cloud radiative kernel used in this study and the code to perform the SW ERFaci decomposition are available at https://zenodo.org/records/13839356 (Duran, 2024). The source code modifications to COSP2 for the new MODIS $r_e$–LWP joint histograms can be found at https://github.com/CFMIP/COSPv2.0/commit/d252f193137b54adff4cc5b8f40604f1832472fa. COSP2 downloading and installation steps can be found at https://github.com/CFMIP/COSPv2.0. MODIS cloud histograms are available from the National Aeronautics and Space Administration (NASA) Level-1 and Atmosphere Archive and Distribution System (https://doi.org/10.5067/MODIS/MCD06COSP_M3_MODIS.062, NASA (2023)). The code for computing APRP was adapted from Zelinka (2023). The code for computing the ISCCP CRK results was adapted from Zelinka (2024).

*Author contributions.* BMD led the research and analysis and wrote the original manuscript draft. CJW designed the conceptualization and developed the methodology, and PM, YQ, MLD, BM, TM, and MD performed simulations. NJL and CJW supervised the work and provided feedback on the analysis. TM contributed discussion that shaped investigation.

*Competing interests.* The authors declare that they have no conflict of interest.

*Acknowledgements.* The authors would like to thank Mark Zelinka, Chris Smith, Duncan Watson-Parris, and Pengcheng Zhang for helpful discussions and feedback. BMD is supported by the National Science Foundation Graduate Research Fellowship under Grant No. DGE-



2038238. CJW received funding from the European Union's Horizon 2020 research and innovation programme under the Marie Skłodowska-
Curie grant agreement No 101019911. PLM and YQ acknowledge support from U.S. Department of Energy, Office of Science, Office of
Biological & Environmental Research, Regional and Global Model Analysis program area and Earth System Model Development pro-
gram area as part of the "Enabling Aerosol-cloud interactions at GLobal convection-permitting scalES (EAGLES)" project (project no.
74358). E3SM simulations are performed using resources of the National Energy Research Scientific Computing Center (NERSC), a U.S.
Department of Energy Office of Science User Facility located at Lawrence Berkeley National Laboratory, operated under Contract No. DE-
AC02-05CH11231, using NERSC awards ALCC-ERCAP0025938, BER-ERCAP0024471, and BER-ERCAP0029295. The Pacific North-
west National Laboratory (PNNL) is operated for the DOE by the Battelle Memorial Institute under Contract DE-AC05-76RL01830. TM
was supported by the JST FOREST Program (grant no. JPMJFR206Y), the Japan Society for the Promotion of Science KAKENHI (grant no.
JP23K13171), MEXT program for the Advanced Studies of Climate Change Projection (SENTAN) (grant no. JP-MXD0722680395), and
the Environment Research and Technology Development Fund (S-20) (grant no. JPMEERF21S12004) of the Environmental Restoration and
Conservation Agency. MIROC numerical simulations were executed with the SX-Aurora TSUBASA supercomputer system of the National
Institute for Environmental Studies, Japan. We would like to acknowledge high-performance computing support from the Derecho system
(doi:10.5065/qx9a-pg09) provided by the NSF National Center for Atmospheric Research (NCAR), sponsored by the National Science Foun-
dation. BM acknowledges support by the U.S. Department of Energy, Office of Biological & Environmental Research, Regional and Global
Model Analysis component of the Earth and Environmental System Modeling Program under Award Number DE-SC0022070 and National
Science Foundation (NSF) IA 1947282; the National Center for Atmospheric Research, which is a major facility sponsored by the NSF under
Cooperative Agreement No. 1852977; and the National Oceanic and Atmospheric Administration under award NA20OAR4310392.



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





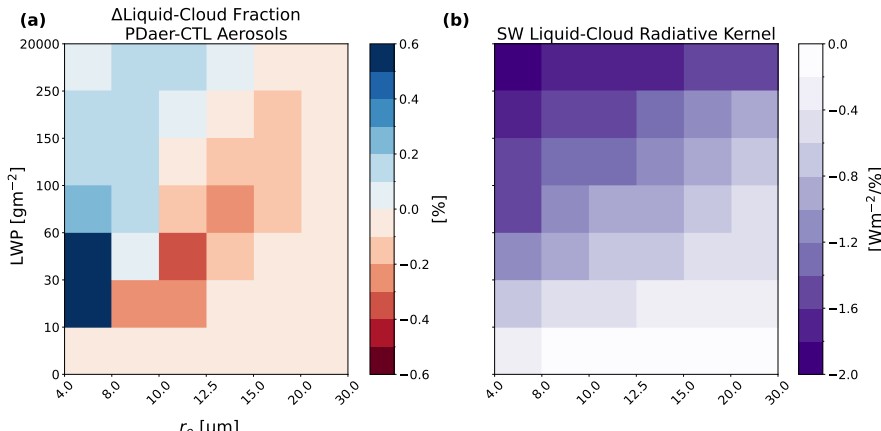

**Figure 1.** Example of the global-mean MODIS cloud fraction histogram and radiative kernel. **(a)** Difference in cloud histograms between the PDaer and CTL simulations for liquid-topped clouds in E3SMv2. **(b)** SW cloud radiative kernel for liquid-topped clouds for E3SMv2 transformed to latitude-longitude space using the CTL clear-sky surface albedo. The cloud histograms and radiative kernels are functions of LWP, $r_\mathrm{e}$, calendar month, latitude, and longitude, but have been spatially and temporally averaged in the example above for presentation.





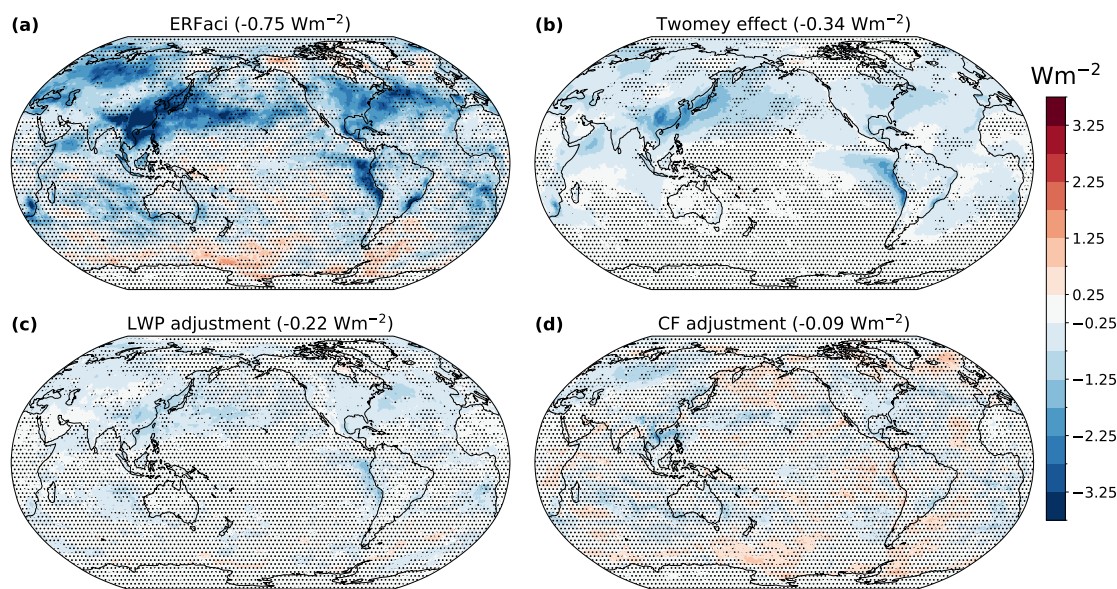

**Figure 2.** Components of SW ERFaci averaged over the five GCMs. **(a)** Total SW ERFaci. Contribution to SW ERFaci from the **(b)** Twomey effect, **(c)** LWP adjustment, and **(d)** CF adjustment. Global-mean values of each component are indicated in parenthesis. Stippling indicates regions where the five models disagree on the sign of SW ERFaci.



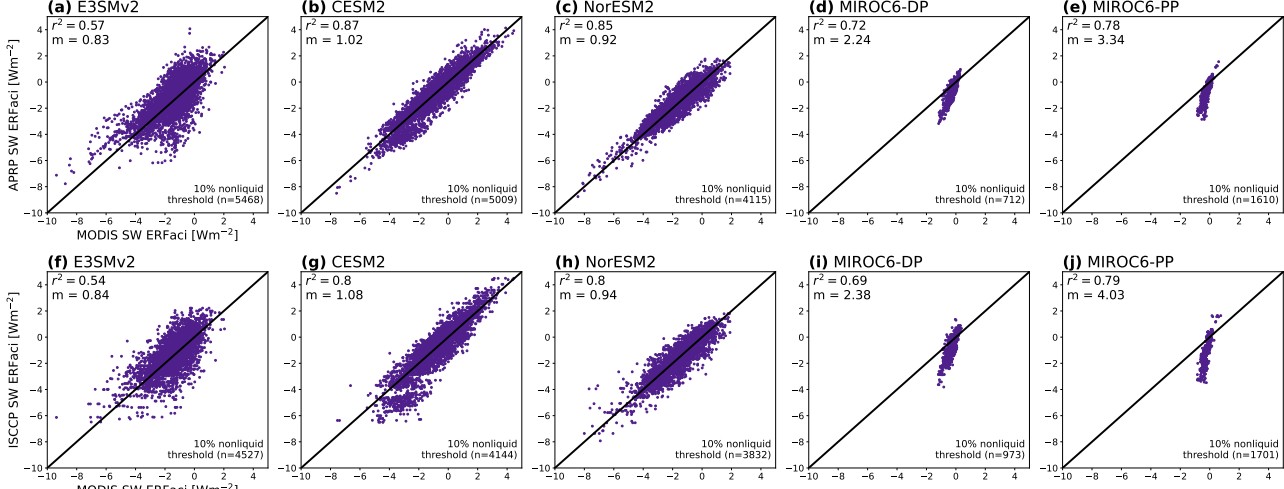

**Figure 3.** Point-by-point comparison of total SW ERFaci computed with the MODIS CRK method presented in this work, the ISCCP CRK method (Zelinka et al., 2012a), and APRP. **(a-e)** Scatter plot of total SW ERFaci diagnosed from the MODIS CRK method and APRP in E3SMv2, CESM2, NorESM2, MIROC6-DP, and MIROC6-PP, respectively. The diagonal line is the one-to-one line. The upper left corner of each panel gives the squared coefficient of determination for the ordinary least-squares linear-regression fit ($r^2$) and the slope of the fit (m). The bottom right corner indicates the number of gridpoints (n) that fall under the threshold of < 10 % annual-mean non-liquid CF and are included in the comparison. **(f-j)** Identical to **(a-e)**, except comparing the total SW ERFaci diagnosed from the MODIS CRK method with that from the ISCCP CRK method.



**Figure 4.** Annual-mean liquid-cloud fraction in MODIS satellite observations and MODIS instrument-simulator output from the nudged historical runs from the five GCMs. LCF in $r_e$–LWP phase space, spatially and temporally averaged over the **(a)** Southeast Pacific (SEP; 30°–10° S, 90°–70° W), **(b)** Southeast Atlantic (SEA; 25°–5° S, 10°W–10° E), **(c)** Northeast Atlantic (10°–30° N, 40°–20° W), **(d)** Australian (35°–15° S, 90°–110° E), and **(e)** Northeast Pacific (15°–35° N, 140°–120° W) regions. Data are temporally averaged over an 11-year period (2003–2014) to ensure overlap. Normalized LCF is calculated by dividing the LCF in each $r_e$–LWP bin by the annual- and global-mean maximum LCF across all of the histograms bins. The vertical black line separates the observational data (OBS, to the left) and the GCM output (to the right). The top right corner of each histogram indicates the total LCF over the region.



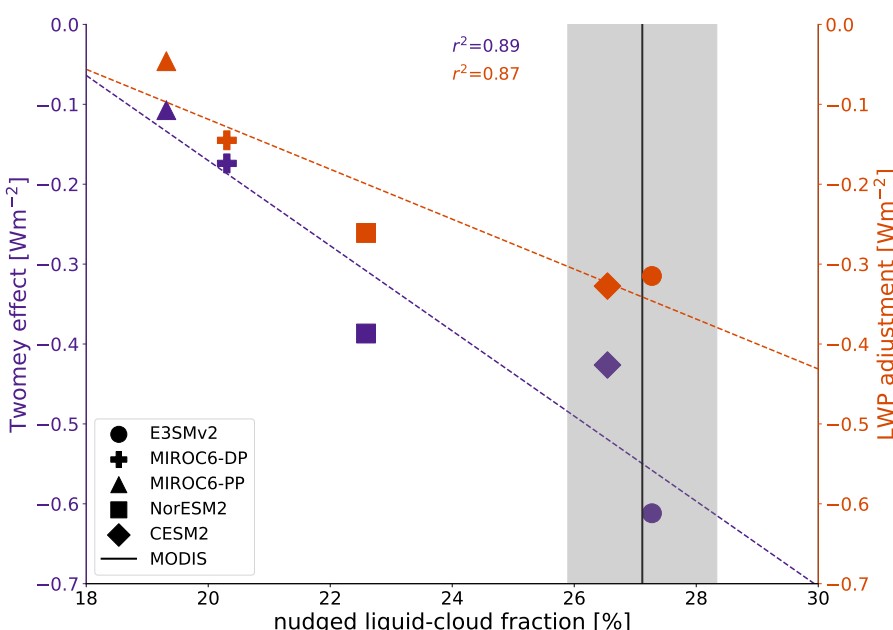

**Figure 5.** Emergent constraints on the global-mean Twomey effect and LWP adjustment. Each shape represents the estimate of the Twomey effect (purple) and the LWP adjustment (orange) for a specific GCM. The vertical black line indicates the MODIS-reported global annual-mean liquid-cloud fraction over the 2003–2014 period. The grey shading represents the 68 % confidence interval ($\pm 1\sigma$ range) for the MODIS global-mean LCF. The values in the top-middle section of the plot give the squared coefficient of determination for the ordinary least-squares linear-regression fit ($r^2$) for each respective fit.



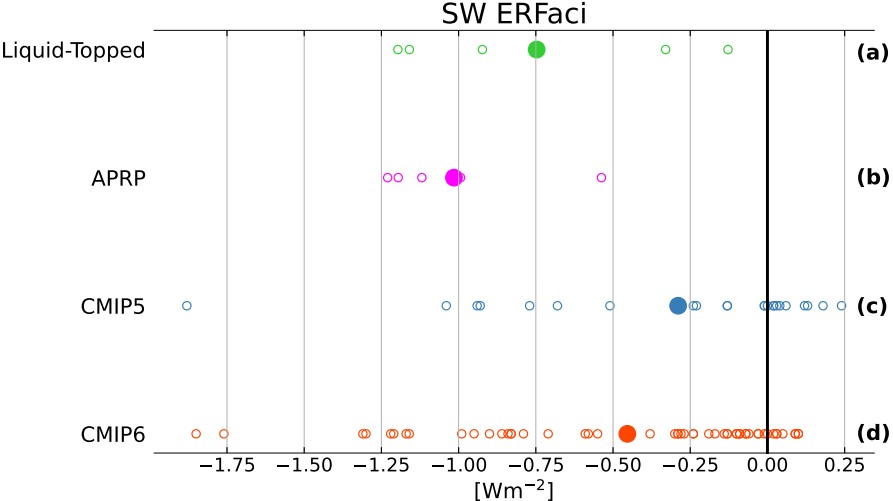

**Figure 6.** A comparison of global-mean SW ERFaci estimates calculated from the new MODIS CRK method and APRP. **(a)** Total SW ERFaci from liquid clouds as diagnosed by our new technique. **(b)** Total SW ERFaci computed using APRP across our ensemble. Total SW ERFaci computed using APRP across an ensemble of **(c)** CMIP5 and **(d)** CMIP6 models from Zelinka et al. (2023). A selection of the models analyzed in this paper (CESM2, MIROC6, NorESM2) are included in the CMIP6 ensemble. Present-day aerosols are defined as year 2000 in our results and year 2014 for CMIP6.





**Table 1.** SW ERFaci, Twomey effect, and adjustments by component [Wm$^{-2}$] for each of the 5 GCM configurations. CF is the obscuration-corrected adjustment. Ensemble-mean values are reported with their 95 % confidence interval ($\pm 2\sigma$ range).

| Model | ERFaci | Twomey | LWP adjustment | CF adjustment |
|---|---|---|---|---|
| E3SMv2 | -1.159 | -0.612 | -0.315 | -0.071 |
| CESM2 | -1.120 | -0.426 | -0.328 | -0.233 |
| NorESM2 | -0.923 | -0.387 | -0.261 | -0.157 |
| MIROC6-DP | -0.330 | -0.174 | -0.145 | -0.017 |
| MIROC6-PP | -0.128 | -0.107 | -0.046 | 0.009 |
| Mean | $-0.747 \pm 0.544$ | $-0.341 \pm 0.226$ | $-0.219 \pm 0.134$ | $-0.095 \pm 0.111$ |



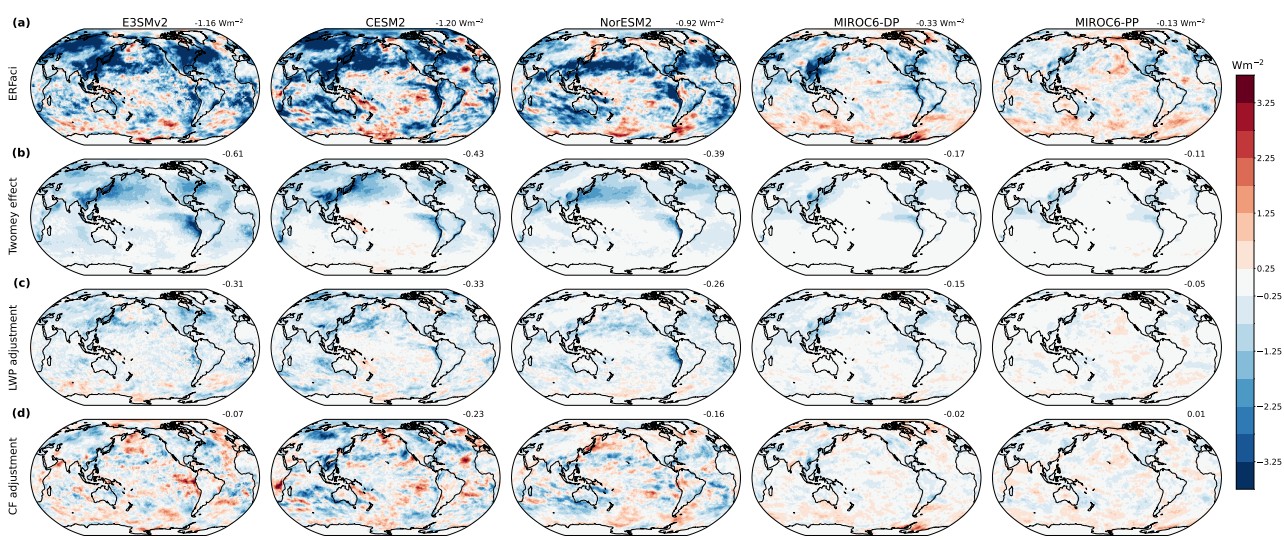

**Figure A1.** Components of SW ERFaci for each of the five GCMs. **(a)** Total SW ERFaci. Contribution to SW ERFaci from the **(b)** Twomey effect, **(c)** LWP adjustment, and **(d)** CF adjustment. From left to right, MODIS CRK decomposition for E3SMv2, CESM2, NorESM2, MIROC6-DP, and MIROC6-PP, respectively. Global-mean values of each component are indicated in the top-right corner of each plot.



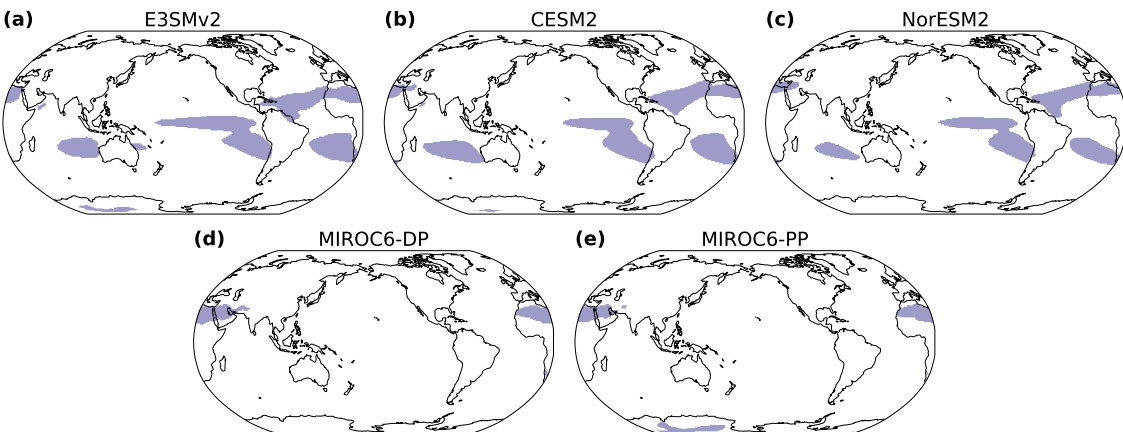

**Figure A2.** Grid points sampled for validation of the new MODIS CRK method with the existing APRP and ISCCP CRK methods for calculating SW ERFaci. Shaded areas denote regions where the annual-mean non-liquid cloud fraction from CTL is less than 10 %. Note that the MIROC6-DP and MIROC6-PP masks do not include the main stratocumulus regions captured in the remaining three GCM masks.



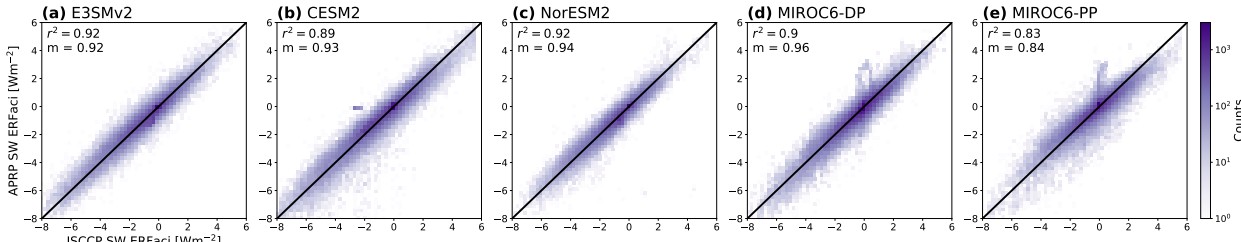

**Figure A3.** Point-by-point comparison of total SW ERFaci computed with the ISCCP CRK method (Zelinka et al., 2012a) and APRP for all gridpoints. **(a-e)** Joint histogram of total SW ERFaci diagnosed from the MODIS CRK method and APRP in E3SMv2, CESM2, NorESM2, MIROC6-DP, and MIROC6-PP, respectively. Purple shading indicates the number of counts on a logarithmic scale. The diagonal line is the one-to-one line. The upper left corner of each panel gives the squared coefficient of determination for the ordinary least-squares linear-regression fit ($r^2$) and the slope of the fit (m).





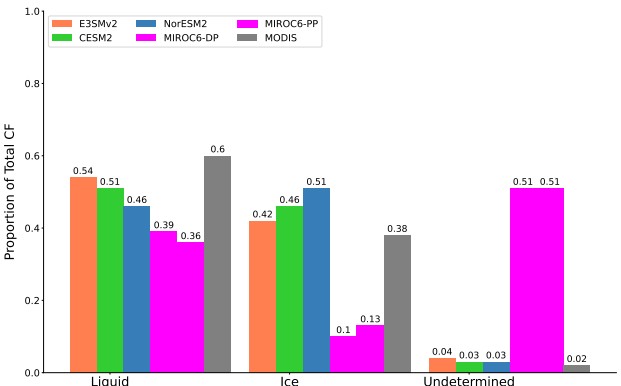

**Figure A4.** Annual-mean total cloud fraction in MODIS satellite observations and MODIS instrument-simulator output from the nudged historical runs from the five GCMs, partitioned by cloud-top phase. The proportion of undetermined clouds is calculated as the difference between the MODIS total cloud fraction and the sum of the MODIS liquid- and ice-phase cloud fraction. Data are temporally averaged over an 11-year period (2003–2014) to ensure overlap. The reported values for MODIS include only fully-cloudy pixels. The proportion of undetermined clouds is similar across all simulations for each model.





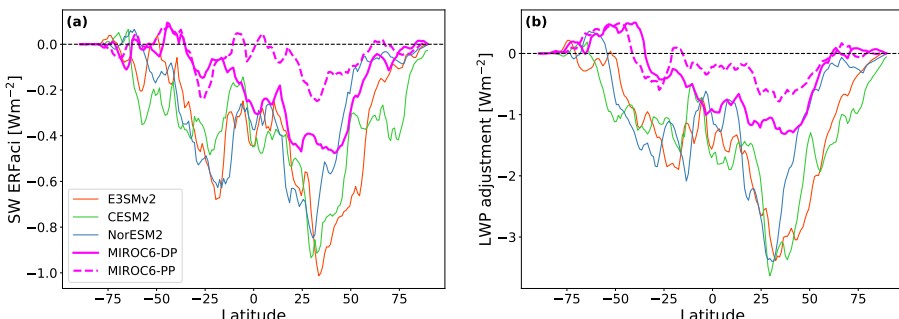

**Figure A5.** Zonal-mean **(a)** total SW ERFaci and **(b)** LWP adjustment, from liquid clouds. The effect of precipitation scheme in MIROC6 is highlighted in the full and dotted lines in purple. Large reductions in MIROC6-PP are observed, especially between 20° and 50° N, consistent with results from Michibata et al. (2019, 2020).



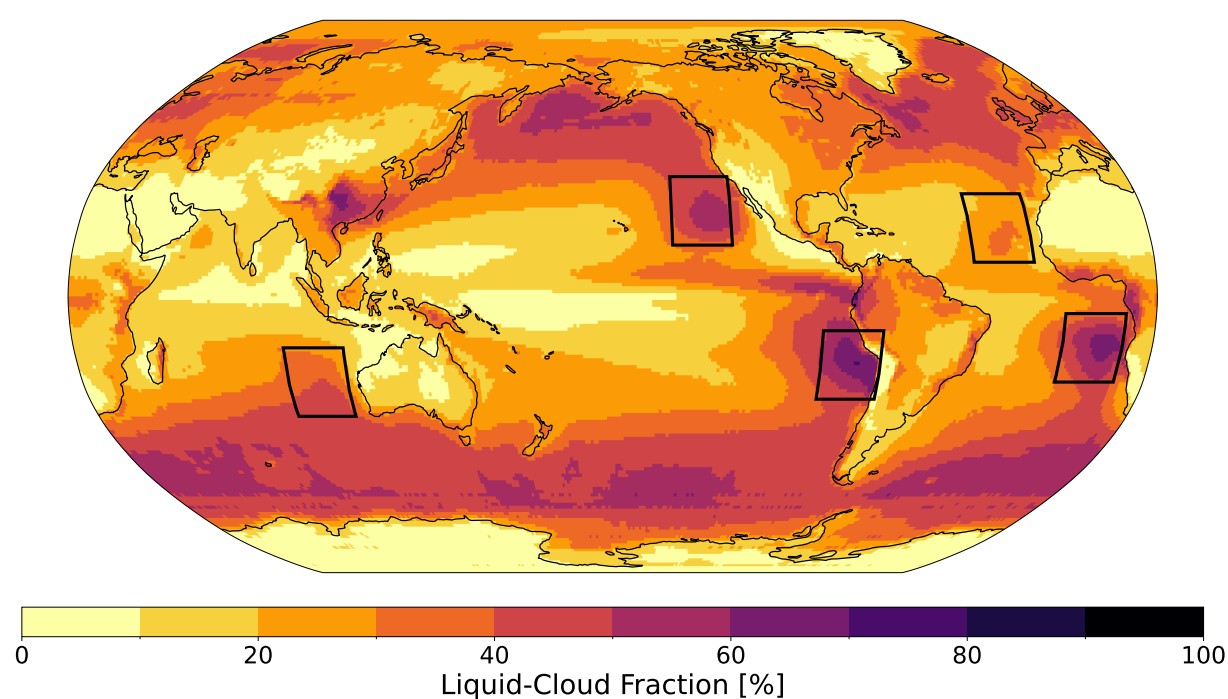

**Figure A6.** Distribution of annual-mean liquid cloud fraction from 11 years (2003–2014) of MODIS observations. Black boxes indicate the five $20° \times 20°$ marine stratocumulus regions used in this study, following Zhang and Feingold (2023).



**Table A1.** Details of the HIST simulations. Each modelling center employed a different nudging procedure. Reanalysis products: MERRA-2: Modern-Era Retrospective Analysis for Research and Applications, Version 2 (Gelaro et al., 2017); ERA5: European Centre for Medium-Range Weather Forecasts (ECMWF) fifth-generation atmospheric reanalysis (Hersbach et al., 2020).

| Model | Variables | Timescale | Products |
|---|---|---|---|
| E3SMv2 | U/V | 6-hourly | MERRA-2 |
| CESM2 | U/V | 1-hourly | ERA5 |
| NorESM2 | U/V/PS | 6-hourly | ERA5 |
| MIROC6-DP | U/V/T | 6-hourly | ERA5 |
| MIROC6-PP | U/V/T | 6-hourly | ERA5 |





**Table A2.** Comparison of estimates of SW ERFaci [Wm$^{-2}$] for each of the 5 GCMs. Our MODIS CRK approach estimates the total SW ER-Faci from liquid clouds ($ACI_{\mathrm{Liq}}^{\mathrm{MODIS}}$), whereas the APRP and ISCCP CRK methods estimate the total SW ERFaci from all clouds ($ACI_{\mathrm{Tot}}^{\mathrm{APRP}}$, $ACI_{\mathrm{Tot}}^{\mathrm{ISCCP}}$). $ACI_{\mathrm{Liq}}^{\mathrm{MODIS}}/ACI_{\mathrm{Tot}}^{\mathrm{X}}$ is the ratio of the SW ERFaci from liquid clouds (MODIS CRK) and the SW ERFaci from all clouds, where X is APRP or ISCCP. $\mathrm{LWP}^{\mathrm{Y}}/ACI_{\mathrm{Liq}}^{\mathrm{Y}}$ is the ratio of the LWP adjustment and SW ERFaci, where Y is MODIS CRK or APRP.

| Model | $ACI_{\mathrm{Liq}}^{\mathrm{MODIS}}$ | $ACI_{\mathrm{Tot}}^{\mathrm{APRP}}$ | $ACI_{\mathrm{Tot}}^{\mathrm{ISCCP}}$ | $\dfrac{ACI_{\mathrm{Liq}}^{\mathrm{MODIS}}}{ACI_{\mathrm{Tot}}^{\mathrm{APRP}}}$ | $\dfrac{ACI_{\mathrm{Liq}}^{\mathrm{MODIS}}}{ACI_{\mathrm{Tot}}^{\mathrm{ISCCP}}}$ | $\dfrac{\mathrm{LWP}^{\mathrm{MODIS}}}{ACI_{\mathrm{Liq}}^{\mathrm{MODIS}}}$ | $\dfrac{\mathrm{LWP}^{\mathrm{APRP}}}{ACI_{\mathrm{Tot}}^{\mathrm{APRP}}}$ |
|---|---|---|---|---|---|---|---|
| E3SMv2 | -1.159 | -1.120 | -1.219 | 0.97 | 0.96 | 0.27 | 0.04 |
| CESM2 | -1.197 | -1.119 | -1.019 | 1.07 | 1.17 | 0.27 | 0.10 |
| NorESM2 | -0.923 | -1.230 | -1.251 | 0.75 | 0.74 | 0.28 | 0.08 |
| MIROC6-DP | -0.330 | -0.993 | -0.888 | 0.33 | 0.37 | 0.44 | 0.10 |
| MIROC6-PP | -0.128 | -0.537 | -0.502 | 0.24 | 0.25 | 0.36 | 0.03 |