# Peer review of "A new method for diagnosing effective radiative forcing from aerosol-cloud interactions in climate models"

_EGUsphere, 2024_

## Referee Comment (RC1)

**Review of Duran, et al. – "A new method for diagnosing effective radiative forcing from aerosol-cloud interactions in climate models"**

This paper describes a technique to calculate aerosol ERF using monthly mean LWP-reff histograms from COSP and radiative kernels. The ERF can also be broken down into contributions from droplet number, LWP and cloud fraction changes. The use of monthly mean histograms means that high frequency bespoke output is not required, hopefully making the technique applicable to more models. The results compare well to other techniques where it is possible to compare them. ERFs and its components are compared between a few different climate models. Evaluation of the histograms from the model those from observations are performed, which is a useful way to assess model performance. The MIRCO6 models perform poorly, which may be due to issues with the cloud phase identification from COSP. An emergent constraint is attempted based on the relationship between the Twomey/LWP adjustment and liquid cloud fraction (constrained by observed cloud fraction).

The paper is very well written and the analysis is sound. I have very few criticisms to make, just a few small points below. The biggest concern is the situation with the MIROC model - it's use in the emergent constraint is dubious given that the results from that model might be being caused by issues identifying the cloud phase from COSP – either that or the model cloud field is very unrealistic. It would be good to determine which is the case.

**Issues**

L32 – "The Twomey effect, or instantaneous radiative forcing (IRFaci)"

- IRF refers to the initial perturbation to the Earth's radiation budget without any tropospheric, land surface or stratospheric adjustments (e.g., Smith, 2020). So the Twomey effect could be an IRF, RF or ERF depending on the climate model set up for how it was quantified. Perhaps you could just use ERF for consistency with the other terms? Or, if the Twomey component is different to the other components (no adjustments vs adjustments) in your breakdown calculation, it would be good to explain why the Twomey component is an IRF and the difference between IRF and ERF. But since this is the introduction and you haven't wrote about what simulations you use yet it might be better to stick to ERF? If so you'd have to correct it throughout the paper.

L62 – It would also be good to mention the technique of Grosvenor (2020) that used a simplified radiative calculation for low liquid clouds to estimate the breakdown of ERFs.

MIROC6 results – the cloud phase identification issue is very important for the evaluation of MIRCO6. It might be good to discuss why there is this issue with MIROC some more. Presumably the fraction of undetermined phase clouds is higher than observed by MODIS, which may tell us something about whether the undetermined categorization is realistic. Is it

possible to evaluate MIROC in stratocumulus regions where it is know that there is little ice – then you could assume all of the model clouds were liquid and evaluate the MIROC cloud fraction again. Maybe this is what you allude to at line 498, but it's not clear whether you are still restricting to only liquid clouds as identified by COSP there. I.e., does MIROC really have such unrealistic cloud fractions, or is it due to the phase issue?

Since the results for this model are based on a very limited spatial sample (due to the cloud phase identification issue) it is perhaps not ideal to use it in the emergent constraint.

**Typos/grammar**

P8, L231 – "The two versions of MIROC6 have the weakest SW ERFaci, because they have less liquid cloud in the control state, so fewer clouds susceptible to aerosol perturbations and weaker aerosol forcing."

This would be better as "The two versions of MIROC6 have the weakest SW ERFaci because they have less liquid cloud in the control state and so fewer clouds susceptible to aerosol perturbations."

P14, L414 – "The relationship between mean state biases and the historical SW ERFaci identified in the previous section suggest" – "suggest" would be better as "suggests".

**References**

Smith, 2020 - https://acp.copernicus.org/articles/20/9591/2020/

Grosvenor, 2020 - https://doi.org/10.5194/acp-20-15681-2020

---

## Author Response (AR1)

**Response to Reviewers**

We thank the two reviewers for their thorough readings of the manuscript and for their helpful and constructive comments. The manuscript has been much improved by addressing their comments. The reviewers' comments are listed below in blue italics and our responses in regular text. Unless otherwise stated, line numbers and page numbers in our responses refer to the revised manuscript. Our proposed changes to the manuscript are surrounded by quotation marks.

**Reviewer 1**
*Comments*

*1. L32 – "The Twomey effect, or instantaneous radiative forcing (IRFaci)"- IRF refers to the initial perturbation to the Earth's radiation budget without any tropospheric, land surface or stratospheric adjustments (e.g., Smith, 2020). So the Twomey effect could be an IRF, RF or ERF depending on the climate model set up for how it was quantified. Perhaps you could just use ERF for consistency with the other terms? Or, if the Twomey component is different to the other components (no adjustments vs adjustments) in your breakdown calculation, it would be good to explain why the Twomey component is an IRF and the difference between IRF and ERF. But since this is the introduction and you haven't wrote about what simulations you use yet it might be better to stick to ERF? If so you'd have to correct it throughout the paper.*

The Twomey effect is not computed any differently than the cloud adjustments in our approach. For clarity, we have removed all instances of "IRFaci" in the manuscript.

*2. L62 – It would also be good to mention the technique of Grosvenor (2020) that used a simplified radiative calculation for low liquid clouds to estimate the breakdown of ERFs.*

We thank the reviewer for this suggestion. We have added the following text to the manuscript (59):
"Grosvenor and Carslaw (2020) developed a method to decompose the SW ERFaci at the surface into components from changes in cloud-droplet number concentration, LWP, and CF. However, their method requires high-frequency output and specialized diagnostics that are not widely available, and to our knowledge the method has only been applied to one GCM. Similarly, Gryspeerdt et al. (2020) presented an approach to break down the SW ERFaci into the Twomey effect, LWP, and CF adjustments; nevertheless, it also relies on aerosol-free diagnostics and high-frequency model output, and decomposes the Twomey effect as a residual term."

*3. MIROC6 results – the cloud phase identification issue is very important for the evaluation of MIRCO6. It might be good to discuss why there is this issue with MIROC some more. Presumably the fraction of undetermined phase clouds is higher than observed by MODIS, which may tell us something about whether the undetermined categorization is realistic.*

We agree the phase-determination bias in MIROC6 is an important issue in the paper. We show in Figure A4 that the fraction of undetermined phase clouds is over 0.5 in MIROC6, which is slightly more than 10x greater than that observed by MODIS. This evidence points to either biases in the model simulation of the cloud field or in the satellite simulator. It is difficult to distinguish between these possibilities, but we have taken some preliminary steps to build our confidence that the satellite simulator has been implemented correctly.

To verify proper implementation of the satellite simulator, and rule out the second case above, we compared liquid cloud fraction (LCF) between the model satellite simulators in the nudged historical runs and MODIS observations over the 2003-2014 period. While MIROC6 has the most biased simulation of liquid clouds of all the GCMs used here, the magnitude of its LCF deficits (in the global-mean) are only 2-3% less than that exhibited by NorESM2. Furthermore, the spatial pattern of LCF in MIROC6 exhibits features seen in observations, such as enhanced LCF along the eastern subtropical ocean basins associated with stratocumulus regions, as well as elevated cloudiness over the North Pacific and the Southern Ocean. The spatial similarity but low biases in liquid clouds compared to observations in MIROC6 alludes to proper implementation of the MODIS simulator and biases in the model cloud field.

[Figure]

*Figure R1:* Comparison of MODIS LCF from the two MIROC6 nudged historical runs and MODIS observations, over the 2003-2014 period. Global-mean LCF is listed in the top-right of each plot. Spatial correlation between model output and observations also provided in the top-right corner. Note the different color scales for the plots, used to highlight the shared spatial

structure of the liquid cloud field, but the liquid cloud amount varies greatly between model and observations.

We have also compared the MODIS simulator to output from the ISCCP satellite simulator. It is not straightforward to compare total CF from the MODIS simulator to either the model native or ISCCP-simulated total CF because the retrieval methods of the MODIS simulator bias the global-mean cloudiness lower. Both simulators output cloud fraction partitioned by cloud optical depth and cloud-top pressure, and the primary differences between the satellites and their retrieval methods are documented in Pincus et al. (2012). The largest differences in the cloud climatologies from each satellite simulator are found for clouds of $\tau$ < 3.6, because the MODIS simulator purposely excludes clouds with $\tau$ < 0.3 to mimic a detection limit. Pincus et al. (2012) noted that the frequency of clouds with 0.3 < $\tau$ < 3.6 cover 33% in the ISCCP record, but only 10% in the MODIS record. This discrepancy is also present in each of the five models analyzed. In the global-mean, all GCMs show close agreement between the model-native and ISCCP-simulated total CF, with the MODIS-simulated total CF ranging between 10-20 % lower than both these values in the global-mean.

[Figure]

*Figure R2:* Three metrics of global-mean total CF (%) from the nudged historical runs compared across the five GCMs: model native, ISCCP-simulated, and MODIS-simulated, from left to right. In the global-mean, the model native and ISCCP-simulated total CF are approximately the same, while the MODIS-simulated total CF is lower.

Again, we emphasize that the lower total CF reported by the MODIS simulator does not indicate a deficiency in the simulator, and is consistent with the observed differences between the ISCCP and MODIS satellites that relate to quality-control decisions made at the pixel-scale retrieval level. Since the MIROC6 models share similar relationships between the three aforementioned measures of total CF compared to the other three GCMs, it appears that MIROC6 has a phase identification problem. Figure R1 shows that MIROC6, just as E3SM and noted in Pincus et al. (2012), captures the climatological differences between the two satellite simulators, primarily the large discrepancy in clouds of $\tau$ < 0.3. The differences across the remainder of the histogram hint at base differences in the simulated model clouds fields between these two GCMs.

[Figure]

*Figure R3:* Difference between ISCCP- and MODIS-simulated total cloud fraction, in cloud-top pressure (CTP) and optical depth ($\tau$) space, for **(a)** E3SM and **(b)** MIROC6-PP. Output from nudged historical runs. Global-mean difference (summed across all histogram bins) is indicated in the top right corner of each plot.

We note that we made no alterations to the cloud-phase identification process in the MODIS simulator, and the code changes introduced to COSP to produce the new MODIS joint histograms were integrated into the five GCMs analyzed without any changes.

Given this evidence, we believe that the MODIS undetermined-phase bias is likely related to either **(1)** too many mixed-phase-topped clouds or **(2)** too many cases where a thin ice cloud overlies a liquid cloud. Determining whether **(1)** or **(2)** contribute most to this bias is beyond the scope of the present work. To summarize, we present evidence that the MODIS simulator was implemented correctly in MIROC6; as such, we believe that the MIROC6 results should remain in the presentation of the constraint, along with a note about our verification. See response to comment 4 for text changes.

*4. Is it possible to evaluate MIROC in stratocumulus regions where it is know that there is little ice – then you could assume all of the model clouds were liquid and evaluate the MIROC cloud fraction again. Maybe this is what you allude to at line 498, but it's not clear whether you are still restricting to only liquid clouds as identified by COSP there. I.e., does MIROC really have such unrealistic cloud fractions, or is it due to the phase issue?*

*Since the results for this model are based on a very limited spatial sample (due to the cloud phase identification issue) it is perhaps not ideal to use it in the emergent constraint.*

We agree that the large biases in MIROC6 (compared to both observations and the other 3 GCMs examined) complicate the emergent constraint approach in Section 4.2. We discuss the limitations of our constraint by discussing the two large caveats to our results in lines 435-450. We choose to include the MIROC6 results in order to be more comprehensive in our results, while also acknowledging their limits extensively.

While we understand the reviewer's wariness of including the MIROC6 results, we believe that the constraint section should remain in the manuscript with a clear discussion of the limitations. We believe this is supported by: **(i)** a strong physical basis underlying our constraint and **(ii)** its presentation in the text as a use case / example of how our new method can be applied in the future. In regard to **(i)**, as outlined in the manuscript, we have physical reason to believe that the Twomey effect from warm clouds should become more negative with higher mean-state liquid cloud fraction, a rationale we believe would hold true with more data. In regard to **(ii)**, we were deliberate in our phrasing of the constraint as a framework, rather than a robust finding. We believe the messaging in the paper is clear that this approach must be tested with a greater sample of models to assess its validity.

We have added the following text to our presentation of the constraint (465):
"Despite the large undetermined phase bias in MIROC6 and our relatively small sample size, our potential emergent constraint possesses a solid physical basis and is presented here to motivate future tests of its robustness."

We have added a note about our verification of the MODIS simulator implementation in MIROC6 (340):
"We note that we have verified that the MODIS simulator is implemented correctly in MIROC6 using model-native and ISCCP-simulated cloud properties. We compared model-native, ISCCP-, and MODIS-simulated metrics of total cloud fraction, and we analyzed climatological differences between the ISCCP and MODIS simulator using Sect. 5 in Pincus et al. (2012) as reference to show that prior expectations of model-native and satellite simulator differences are reproduced in MIROC6, as in the other three GCMs in this study (not shown)." Furthermore, we have revised the text to state "*potential* emergent constraint" in the majority of our discussion.

*Typos/grammar*

*L231 – "The two versions of MIROC6 have the weakest SW ERFaci, because they have less liquid cloud in the control state, so fewer clouds susceptible to aerosol perturbations and weaker aerosol forcing."*
*This would be better as "The two versions of MIROC6 have the weakest SW ERFaci because they have less liquid cloud in the control state and so fewer clouds susceptible to aerosol perturbations."*
*L414 – "The relationship between mean state biases and the historical SW ERFaci identified in the previous section suggest" – "suggest" would be better as "suggests".*

Both typo/grammar comments have been corrected, thanks.

**Reviewer #2**
*Major Comments:*
*1. My only real comment is that the method doesn't quite decompose linearly. Summing the components from each model in Table 1 underestimates the ERFaci for the three non-MIROC models by more than 10%. I suggest that the residual term is given the same level of prominence in this table and the text. In the opinion of the authors, is this a non-linearity due to second order cross-terms, or could there be other aspects of the ERFaci that we don't or can't evaluate?*

The apparent lack of linearity of the decomposition is a function of how the values in Table 1 were reported. As noted in lines 212-218, we perform a correction to the cloud fraction (CF) adjustment term of the decomposition to account for obscuration effects from changes in the nonliquid cloud fraction. This correction to the CF adjustment removes any contamination from free-tropospheric cloud changes (Scott et al., 2020; Zelinka et al., 2024). In Table 1, we initially reported only the obscuration-corrected CF adjustment, rather than the raw CF adjustment. The linearity of the decomposition can be seen when adding the uncorrected CF adjustment to the Twomey effect and LWP adjustment, with a small residual remaining. For each individual model, the residual term is about 5% of the diagnosed SW ERFaci (in the global-mean), so cross-terms are largely negligible.

To provide more clarity on this and more emphasis on the residual term resulting from our decomposition, we have added the following columns to Table 1: "CF adj.", "Residual." Further, we have changed the existing column "CF adjustment" to "CF adj. (unobscured)". Additionally, we have added the two components mentioned above as new rows to Figure A1 to show the raw CF adjustment and residual term of the decomposition for each of the individual models. We have also added the following text to the Results section (305):

"In four of the five models, the residual term accounts for roughly 5 % of the total SW ERFaci (in the global-mean), with the exception of E3SMv2 (7%), indicating that ERFaci can be decomposed linearly."

Furthermore, we have adjusted all references of SW ERFaci to account for this correction. This includes values reported in figures, tables, and text.

*Prior reported ERFaci values:*
    ERFaci = Twomey + LWP adj. + CF adj. (raw) + residual
*New reported ERFaci values:*
    ERFaci = Twomey + LWP adj. + CF adj. (obscuration-corrected) + residual

*Minor comments:*
*Line 32: just clarify that this is the instantaneous forcing due to aerosol-cloud interactions.*
We have removed all references to IRFaci in this manuscript, following the suggestions from
Referee #1.

*Lines 128-132: could you clarify? >70% liquid means classified as liquid, <30% liquid means*
*classified as ice, and 30-70% liquid means undetermined. Did you also mean 0.3 to 0.7 optical*
*thickness in line 131, or is this a coincidence?*

MODIS determines cloud thermodynamic phase from the cloud condensate between the
retrieved cloud top and one optical depth unit below, for reasons detailed in King et al., 2010. If
extinction in this range of optical depth is between 30-70% liquid, then the cloud is classified as
undetermined phase.

For instance, if an ice cloud of optical depth 0.4 overlies a liquid cloud of optical depth of 10, the
MODIS simulator will consider the 0.4 optical depth units of ice cloud and the uppermost 0.6
optical depth units of liquid cloud. Since the extinction in the subset of cloud condensate is 40%
ice and 60% liquid, MODIS will classify the column as an undetermined phase cloud. In general,
this will happen for such a situation when the ice cloud has optical thickness between 0.3 and
0.7.

**References**:
King, M. D., Platnick, S.,Wind, G., Arnold, G. T., and Dominguez, R. T.: Remote sensing of
radiative and microphysical properties of clouds during TC 4 : Results from MAS, MASTER,
MODIS, and MISR, Journal of Geophysical Research: Atmospheres, 115, 2009JD013 277,
https://doi.org/10.1029/2009JD013277, 2010.
Pincus, R., Platnick, S., Ackerman, S. A., Hemler, R. S., and Patrick Hofmann, R. J.:
Reconciling Simulated and Observed Views of Clouds: MODIS, ISCCP, and the Limits of
Instrument Simulators, Journal of Climate, 25, 4699–4720,
https://doi.org/10.1175/JCLI-D-11-00267.1, 720 2012.
Scott, R. C.,Myers, T. A., Norris, J. R., Zelinka,M. D., Klein, S. A., Sun,M., and Doelling, D. R.:
Observed Sensitivity of Low-Cloud Radiative Effects to Meteorological Perturbations over the
Global Oceans, Journal of Climate, 33, 7717 – 7734, https://doi.org/10.1175/JCLID-19-1028.1,
2020.
Zelinka, M. D., Smith, C. J., Qin, Y., and Taylor, K. E.: Comparison of methods to estimate
aerosol effective radiative forcings in climate models, Atmospheric Chemistry and Physics, 23,
8879–8898, https://doi.org/10.5194/acp-23-8879-2023, 2023.